# Reward modulates the effect of visual cortical microstimulation on perceptual decisions

Nela Cicmil[1]*, Bruce G Cumming[2], Andrew J Parker[1], Kristine Krug[1]

[1]Department of Physiology, Anatomy & Genetics, University of Oxford, Oxford, United Kingdom; [2]Lab of Sensorimotor Research, National Eye Institute, National Institutes of Health, Bethesda, United States

**Abstract** Effective perceptual decisions rely upon combining sensory information with knowledge of the rewards available for different choices. However, it is not known where reward signals interact with the multiple stages of the perceptual decision-making pathway and by what mechanisms this may occur. We combined electrical microstimulation of functionally specific groups of neurons in visual area V5/MT with performance-contingent reward manipulation, while monkeys performed a visual discrimination task. Microstimulation was less effective in shifting perceptual choices towards the stimulus preferences of the stimulated neurons when available reward was larger. Psychophysical control experiments showed this result was not explained by a selective change in response strategy on microstimulated trials. A bounded accumulation decision model, applied to analyse behavioural performance, revealed that the interaction of expected reward with microstimulation can be explained if expected reward modulates a sensory representation stage of perceptual decision-making, in addition to the better-known effects at the integration stage.

*For correspondence:
nela.cicmil@dpag.ox.ac.uk

Competing interests: The authors declare that no competing interests exist.

## Introduction

A central issue for models of perceptual decision-making is how information about the value associated with different choices is combined with sensory information to influence the final decision. Balancing sensory evidence with the prospect of reward must take place on a case-by-case basis for every trial in which a decision is made. When sensory information is used simply to signal the availability of reward in value-based decisions, the activity of lateral intraparietal (LIP) neurons correlates with the relative reward an animal can expect for a particular choice (*Platt and Glimcher, 1999*; *Sugrue et al., 2004*). In perceptual decision-making, however, making a correct decision requires the discrimination of one version of the stimulus from another. Experimentally, different perceptual choices can be arranged to receive greater or lesser reward, contingent upon a correct decision. Such manipulations affect perceptual decision-making, for example by improving performance or biasing decisions (e.g., *Summerfield and Koechlin 2010*; *Weil et al. 2010*). However, the neural mechanisms by which reward signals influence perceptual decision-making are not fully understood.

Perceptual decision-making has been successfully modelled as a bounded evidence-accumulation process in which sensory evidence about the visual stimulus is represented in sensory cortex and then integrated over time in sensorimotor structures such as lateral intraparietal cortex (area LIP) (*Shadlen and Newsome, 2001*; *Roitman and Shadlen, 2002*; *Mazurek et al., 2003*; *Huk and Shadlen, 2005*; *Gold and Shadlen, 2007*) (*Figure 1*). Previous human neuroimaging and animal electrophysiology studies found that when perceptual decision-making is modelled in this way, information about reward is reflected only at the integration stage, represented in sensorimotor structures such as area LIP (*Rorie et al., 2010*; *Summerfield and Koechlin, 2010*; *Mulder et al., 2012*). But does reward

**eLife digest** Identifying how an object is moving in three-dimensional (3D) space depends upon a brain region known as V5/MT. The neurons that make up area V5/MT form groups that each have a 'preference' for a particular direction of movement and a particular 3D depth. If a group of neurons detects its preferred direction of movement and 3D depth, it will become highly active. The brain can assess which groups of neurons are active, in a process known as integration. This information can then be used to work out the object's movement in space.

The process of integration can be influenced by whether a rewarding outcome is expected to result from identifying the 3D movement correctly. This allows the brain to increase its likelihood of success in situations where a large reward is on offer. Until now, it was thought that the activity in area V5/MT, which takes place before integration, was not affected by the likelihood of receiving a reward.

As well as being 'naturally' stimulated by moving objects, the V5/MT neurons can also be 'artificially' activated by a technique called microstimulation, which uses a tiny electrode to electrically stimulate groups of neurons. Microstimulation can bias visual perception towards the movement and 3D depth 'preference' of the artificially activated neurons. If the V5/MT neurons do receive information about potential rewards from other areas of the brain, we would expect rewards to affect naturally and artificially stimulated neural activity in different ways. On the other hand, if the V5/MT neurons do not receive any information about reward, then it will not matter whether their activity is natural or artificial; the signal that they produce will be the same.

Cicmil et al. gave two monkeys a task in which they could receive rewards for correctly identifying a three-dimensional cylinder's direction of rotation, and applied microstimulation to specific groups of V5/MT neurons on some of the trials. When a larger reward was available, microstimulation was less able to bias the monkeys' choices about the rotation direction of the 3D cylinders.

Overall, Cicmil et al.'s results suggest that the V5/MT neurons are able to incorporate information about reward, before integration occurs. The next step will be to record the activity of area V5/MT to investigate exactly how this happens.

simply affect the accumulation of sensory evidence and the motor response, or does it also affect neuronal representations of the sensory stimulus, for example in visual cortical area V5/MT (*Figure 1*)?

There is some indication that expected reward can affect sensory representations. Blood-oxygen level dependent (BOLD) signals in sensory cortex are modulated by reward probability, size and delivery (*Pleger et al., 2008*; *Serences and Saproo, 2010*; *Weil et al., 2010*). However, the BOLD signal is an indirect measure of neural activity that averages the responses of many neurons, including those that are excited by the sensory stimulus and those that are inhibited by it (*Logothetis et al., 2001*; *Logothetis and Wandell, 2004*), and may be differentially affected by modulatory signals in comparison to neuronal spiking (*Boynton, 2011*). A stronger indication comes from a study in which neuronal responses from area V1 were electrophysiologically recorded while animals performed a saccadic curve-tracing task (*Stanisor et al., 2013*). This study showed that V1 activity was predicted by the reward value of a stimulus relative to other stimuli, implying that reward signals have the potential to reach the earliest stages of sensory cortical processing (*Figure 1*).

Experimental interference with the activity of sensory cortex may directly induce changes in the subject's behaviour or their perceptual report, and provides a tool with which to dissect the perceptual decision-making pathway. Such interventions provide strong evidence for a causal role of that region in generating that behaviour or report (*Parker and Newsome, 1998*). Electrical microstimulation of small populations of neurons in relevant cortical regions, combined with sensory stimulation, can produce reliable changes in perceptual reports, which are specific to the stimulus preferences of the neurons at the microstimulated site (*Salzman et al., 1990*, *1992*). When the stimulated neurons encode combinations of visual cues, microstimulation adds a signal that is specific to that combination, as found in cortical area V5/MT for visual motion and binocular disparity (*Krug et al., 2013*). Using microstimulation to bias perceptual choices under different reward conditions is a novel approach with which to explore the interaction between expected reward and sensory signals that contribute directly to perceptual decisions.

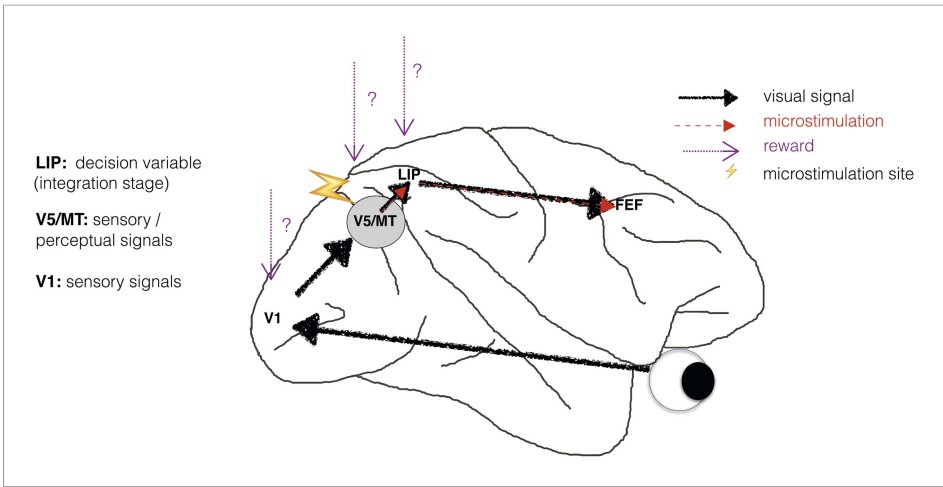

**Figure 1**. Schematic illustration of a visual perceptual decision-making pathway with V5/MT microstimulation and reward. Primary visual cortex (area V1) comprises the initial stage of visual information processing but neurons here can respond to visual sensory signals that do not reach perception (*Cumming and Parker, 1997*). On the other hand, activity of motion- and disparity-selective neurons in visual area V5/MT have been closely linked to animals' subjective perception during discrimination of structure-from-motion (SFM) visual stimuli (*Dodd et al., 2001*). In the present study, we artificially activate motion- and disparity-selective neurons in visual area V5/MT with electrical microstimulation. Microstimulation biases perceptual choices towards visual interpretations that match the tuning of the stimulated neurons (*Krug et al., 2013*). In visual discrimination tasks, sensory evidence is integrated over time into a decision variable (DV), represented in the activity of neurons in lateral intraparietal cortex (area LIP; see *Gold and Shadlen 2007* for review). Since visually evoked and electrically evoked signals both influence behaviour, they are presumably integrated together to influence the DV. The results of evidence integration also affect activity in the frontal eye fields (FEF) that represent the planning of eye movements (saccades), which animals use to indicate their perceptual decision in the discrimination task (*Figure 2*). Previous studies have established an influence of reward on neural representations in sensorimotor regions such as area LIP (*Platt and Glimcher, 1999*; *Sugrue et al., 2004*; *Rorie et al., 2010*). However, it is not known whether reward can also affect sensory or perceptual signals represented in visual cortex during perceptual decision-making.

We investigated the interaction between electrical microstimulation of visual cortical area V5/MT and available reward size for a correct choice during visual discrimination of a structure-from-motion (SFM) stimulus. The available reward size for a correct choice was varied according to animals' preceding behavioural performance. When electrical microstimulation is applied in a sensory area during a perceptual decision task, any effect of reward that acts exclusively at the integration stage, by which time visually and electrically evoked sensory signals have been combined (*Figure 1*), would not differentially affect visually evoked and electrically evoked signals. On the other hand, reward could have a differential effect on neurons that have been specifically excited by either visual or electrical microstimulation. We found that the size of the microstimulation effect on perceptual choices was modulated by the size of the available reward for a correct choice. In the context of a simplified bounded evidence-accumulation model, these results were best explained by an effect of reward on neuronal representations of sensory evidence, in addition to an effect on the integration stage.

## Results

We trained two macaque monkeys (Fle & Ica) to discriminate the direction of rotation of a transparent cylinder defined by SFM in a random-dot kinematogram. This stimulus portrays two transparent surfaces of moving dots, with the dots on one surface moving in the opposite direction from the dots on the other surface. This pattern of motion generates a percept of a rotating cylinder with a compelling sensation of 3D depth (*Treue et al., 1991*) (*Video 1*). Adding binocular disparities to the dots defines the depth order of the two dot surfaces and specifies the direction of rotation of the cylinder to be clockwise (CW) or counter-clockwise (CCW) as viewed from above. If both dot surfaces

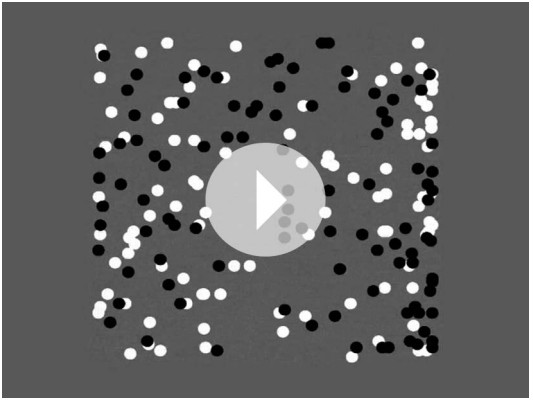

**Video 1.** The structure-from-motion rotating cylinder stimulus. An illustration of the SFM rotating cylinder stimulus used in the study. This example video shows the ambiguously rotating zero-disparity version of the cylinder, for which perception of the direction of rotation is bistable.

have zero disparity, then the depth order of the stimulus is ambiguous and the perception is bistable (*Nawrot and Blake, 1989*; *Parker and Krug, 2003*). Cylinders were presented at a range of binocular disparities, measured in degrees of visual angle (°), in a pseudo-randomized order (*Dodd et al., 2001*). At the end of each 2000 ms trial, animals indicated their decision about cylinder rotation with a saccadic eye movement to the appropriate target (*Figure 2*). When the binocular disparity objectively defined an unambiguous rotation, the animal was rewarded for a correct choice of rotation direction. When there was zero binocular disparity, there was no correct answer and choices were rewarded randomly on 50% of these trials. Electrical microstimulation of motion- and disparity-selective multi-units (MU) in visual cortical area V5/MT was applied during visual stimulus presentation in a randomly selected half of the trials. Area V5/MT has been shown to have a specific role in 3D depth processing (*Dodd et al., 2001*; *Krug et al., 2004*; *Krug and Parker, 2011*), and to contain specialised intrinsic connectivity suitable for combining visual depth and motion signals (*Ahmed et al., 2012*), relevant for the cylinder stimulus. As previously reported, electrical microstimulation of area V5/MT biases animals' choices towards the cylinder rotation direction preferred (PREF) by the MU ($\chi^2$ likelihood-ratio test of nested models *Equations 1a,b*, p < 0.05 in 28 out of 48 sites; Fle: 13/20 sites; Ica: 15/28 sites) (*Krug et al., 2013*). When combined with simultaneous visual stimulation, these artificially induced signals are effectively integrated into perceptual decisions and consistently influence behaviour in this visual task.

The two-alternative forced-choice paradigm dynamically updated the reward contingencies for the animal depending on whether the animal's response was correct or incorrect. When the choice on the immediately preceding trial was correct, the fluid reward volume for the current trial was increased in two stages up to a maximal quantity (*Figure 2*). The largest reward size was therefore available for longer sequences of consecutively correct choices, to incentivize animals towards better performance. 'Large' reward was twice the average volume of 'small' rewards. All trials with reward size less than maximal were classified as 'small reward', as there were too few trials with the intermediate reward size to allow meaningful further division by size. We provided rewards for correct decisions about the visual stimulus regardless of rotation direction, so changes in available reward size did not bias perceptual decisions towards one direction or another.

## Large reward trials are associated with better performance

In agreement with the existing literature about reward and perceptual decisions, available reward size affected perceptual choices. Trials from psychophysical (non-microstimulated) experimental blocks were separated according to available reward size and pooled across blocks and animals. There was a significant interaction between the slope of the fitted psychometric function and available reward size, indicating better performance in large reward trials ($\chi^2$ likelihood-ratio test of nested models *Equations 2a,b*, 'Materials and methods': p = 0.001; *Figure 3*). Since the reward schedule was dependent on performance history, fluctuations in animals' performance over the course of a block could have contributed to a correlation between large reward trials and high performance. However, the control analyses for performance fluctuations (see below) show that such fluctuations do not explain the main results of the study.

## Bounded accumulation can model interactions between reward and perceptual decision-making

It is not well understood at which stages of cortical processing for perceptual decision-making neural signals for reward are integrated (*Figure 1*). However, the neural mechanisms of perceptual

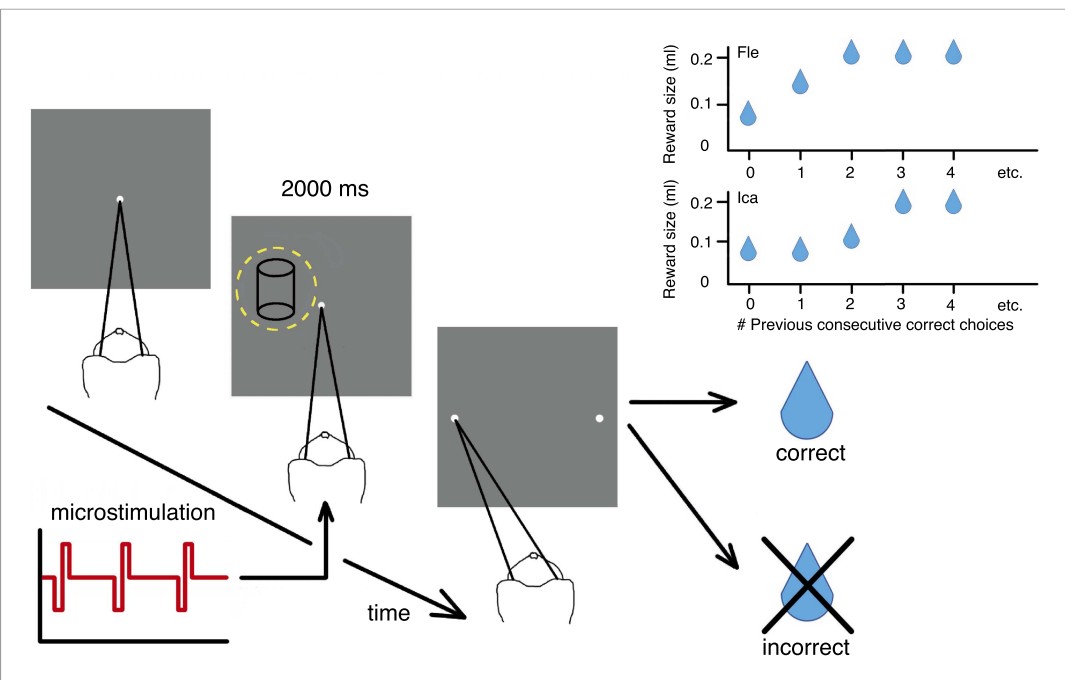

**Figure 2**. Cylinder discrimination task with electrical microstimulation and variable reward schedules. At the beginning of each trial, the monkey looked at the fixation point. The visual stimulus was then presented for 2000 ms, within the receptive field (RF) of the V5/MT multi-unit site selected for microstimulation (yellow dashed circle indicates RF location). Upon stimulus offset, two choice targets appeared. The monkey indicated its perceptual decision with an eye movement to one of the choice targets (left for clockwise [CW] rotation and right for counter-clockwise [CCW]). If the response was correct, the animal received a fluid reward; if incorrect, it received no reward and a time-out. On 50% of trials (pseudo-randomly chosen), electrical microstimulation was added to the V5/MT multi-unit site during visual stimulus presentation. Inset panels: Fluid reward volume increased in two steps up to a maximum depending upon the animal's performance history, that is, the number of immediately preceding consecutive correct choices. Reward size increased more quickly for Fle than Ica as a function of consecutive correct responses, but for both animals we categorized the trials into two conditions: maximal reward size ('large') and sub-maximal reward size ('small'). The average size of sub-maximal rewards was half the maximal reward size for both animals.

decision-making per se have been modelled successfully by a bounded evidence-accumulation process in which representations of sensory information are noisily integrated over time towards decision bounds that represent the possible perceptual outcomes (*Gold and Shadlen, 2007*; *Ratcliff and McKoon, 2008*) (*Figure 4A*). Considering the possible effects of reward on different parameters in the bounded accumulation model can help us to interpret how expected reward influences perceptual decisions.

In the bounded accumulation model, sensory evidence about the cylinder stimulus is represented as the difference in activity of two pools of sensory neurons, where the conjoint motion and disparity tuning of each pool represents one of the two possible interpretations of cylinder rotation direction (*Mazurek et al., 2003*; *Palmer et al., 2005*; *Gold and Shadlen, 2007*; *Krug et al., 2013*) (*Figure 4B*). Sensory evidence is integrated over time to influence the drift-diffusion of the decision variable (DV) (*Huk and Shadlen, 2005*). The DV starts at a particular distance from the two decision bounds and, during stimulus viewing, drifts stochastically towards one or other decision bound at a rate proportional to the sensory evidence favouring the prospective choices. Upon stimulus offset, the animal makes a saccade towards the target that corresponds to its perceptual choice, which depends upon either the first bound that was crossed or, if no bound was crossed, the bound nearer to the DV final position (*Kiani et al., 2008*; *Fetsch et al., 2014*) (*Figure 4B*). The distance between the DV starting point and the decision bounds predicts the reaction time distribution in a reaction time task; in the present fixed viewing duration task, it offers predictions of decision times that are shorter than

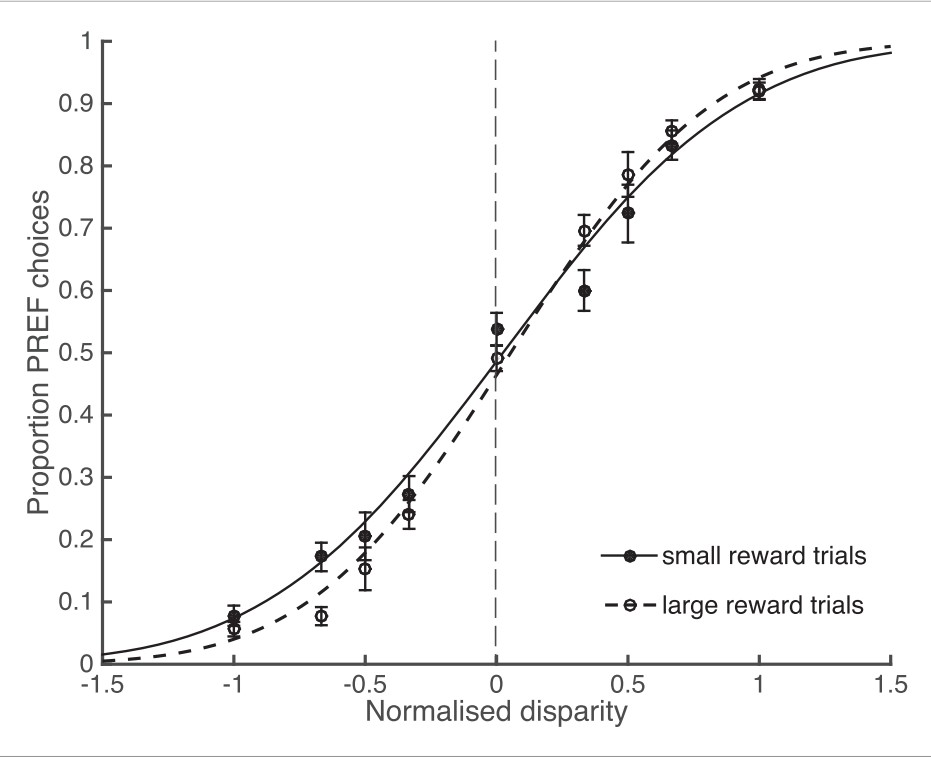

**Figure 3**. Large reward size is associated with improved performance. Animals performed two blocks of the visual discrimination task at each site, prior to the introduction of electrical microstimulation. Trials were separated according to available reward size (small or large) and pooled across all sites over both animals. Psychometric functions are fit with cumulative Gaussians. There is a significant interaction between reward size and steepness of the slope (s.d. of the fitted function; $\chi^2$ likelihood-ratio test of nested models **Equations 2a,b**: p = 0.001). For large reward trials the slope is steeper, indicating better performance accuracy. Error bars show the standard error of the mean (s.e.m.).

the stimulus duration and, critically for our data, it predicts performance accuracy (*Kiani et al., 2008*; *Fetsch et al., 2014*).

According to the bounded accumulation model, an increase in either the stimulus sensitivity (represented by parameter *k*) or the distance between the DV starting point and the bounds (represented by parameter *B*, *Equation 3a*, 'Materials and methods'; *Figure 4B*) can improve performance (*Figure 4C*; compare to *Figure 3*). This is because increasing the perceptual sensitivity to the stimulus allows smaller stimulus disparity signals to have a greater effect on the perceptual decision; on the other hand, increasing the diffusion distance to the decision bounds decreases the relative impact of spurious noise on the final decision (*Bogacz et al., 2006*; *Ratcliff and McKoon, 2008*). The contributions of these parameters cannot be differentiated based upon these perceptual choice data alone because they both improve performance. However, these parameters make different predictions about what should happen to visual cortical microstimulation when performance improves. Electrical microstimulation biases perceptual decisions towards the PREF direction of the stimulated MU site (*Figure 4D*; *Krug et al., 2013*). If performance in large reward trials improves as a result of an increase in *B*, this will not affect the size of the microstimulation shift, because *B* acts at an integration stage that is positioned after the location at which visually evoked and electrically evoked signals have been combined (*Figure 4E*; see also *Figure 1*). If performance improves as a result of an increase in *k*, this will decrease the relative influence of microstimulation on perceptual choices (illustrated by a smaller horizontal shift between microstimulated and non-microstimulated psychometric functions) because *k* acts specifically on visually evoked sensory representations before they are combined with electrical microstimulation signals (*Figure 4F*). Therefore, the insertion of the electrical microstimulation signal at the level of representation of sensory evidence (area V5/MT) allows us to separate the effects of parameters *k* and *B* according to this model.

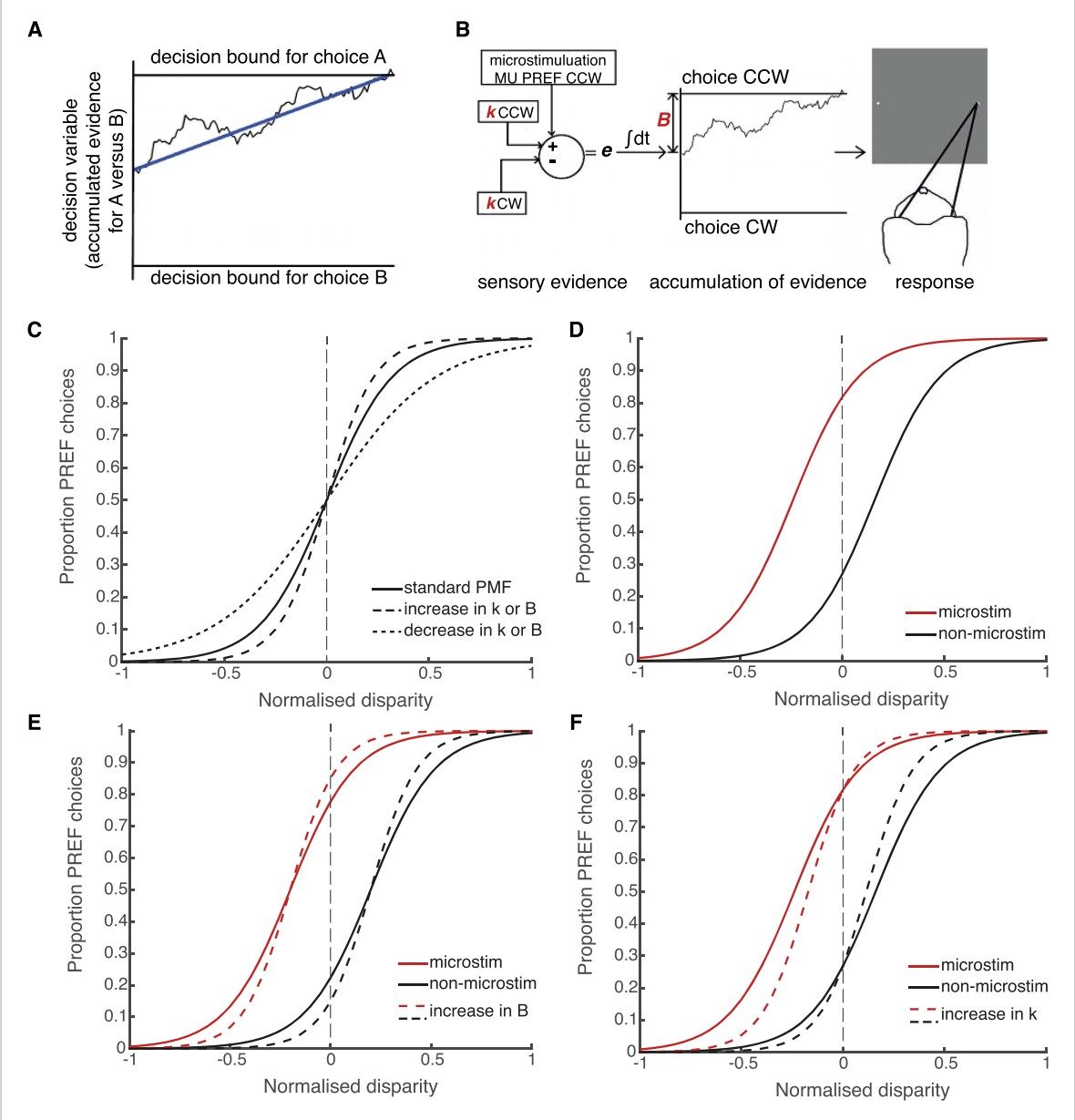

**Figure 4**. Schematic illustration of possible interactions between microstimulation, reward and the parameters of the bounded accumulation model of perceptual decision-making. (**A**) Momentary evidence, in favour of choice 'A' over choice 'B', is accumulated over time. The time-varying accumulation of evidence is termed the *decision variable* (DV; black line). (**B**) The bounded accumulation model applied to the cylinder task with microstimulation. The momentary evidence (*e*) is the difference in activity between two neuronal pools, one selective for CW rotation and one for CCW rotation. Parameter *k* represents the sensitivity of these sensory representations to the cylinder disparity *C* (***Palmer et al., 2005***). In each trial, the DV follows a stochastic 'drift-diffusion' path with an average drift rate dependent on the mean of *e* = *kC*. Parameter *B* represents the distances to the CW and CCW decision bounds, which are assumed to be equal. Microstimulation contributes to *e* as additional evidence in favour of the sensory preference of the stimulated V5/MT multi-unit site (CCW in this example). The perceptual decision depends on which bound was first reached, or, if neither is reached, which bound the DV is nearest to at the end of stimulus viewing. (**C**) Model simulations illustrate that the slope of the psychometric function, fitted by a simple logistic model of bounded accumulation (***Equation 3a***), can be affected by changes in either parameter *k* or *B*; for example, an increase in either parameter will steepen the slope, indicating improved performance. (**D**) Model simulations illustrate how insertion of additional sensory evidence by electrical microstimulation in visual cortex biases perceptual choices toward the preferred (PREF) cylinder disparity of the stimulated multi-unit (MU), which is revealed as a shift of the psychometric function. (**E**) An increase in parameter *B* steepens the slope of the psychometric functions, but does not change the size of microstimulation shift. This is because by the evidence-accumulation stage, visually evoked and electrically evoked signals are combined and cannot be differentially affected (see panel B). (**F**) An increase in parameter *k* also steepens the slope of the psychometric function. However, improved perceptual sensitivity to the stimulus affects only the visually-evoked sensory representations (see panel B), so the relative contribution of the electrically-evoked signals is decreased, which results in a decrease in the microstimulation-induced shift of the psychometric function.

# Expected reward size modulates the effect of microstimulation on perceptual decisions

We fit the bounded accumulation model of decision-making (*Equation 3a*, 'Materials and methods') to behavioural choice data combined across significant PREF microstimulation sites (sites with a significant microstimulation effect in their PREF cylinder rotation direction), separately for each animal (*Figure 5*). Prior to pooling, cylinder disparity values were normalised by the maximum disparity from the range used at each site. The size of the microstimulation effect - the horizontal shift between microstimulated and non-microstimulated psychometric functions—was decreased in large reward trials (dashed lines) compared to small reward trials (smooth lines), implicating a change in stimulus sensitivity (parameter *k*) by reward (see *Figure 4F*). Using a $\chi^2$ likelihood-ratio test, we compared the goodness-of-fit of the full bounded-accumulation model in which all parameters were allowed to vary by reward (*Equation 3a*, 'Materials and methods') to a nested model in which the parameter in question (either *k* or *B*) was frozen, that is, not allowed to vary by reward (*Equations 3b,c*, respectively).

For both animals, the overall best-fitting model was the full model in which both parameters *B* and *k* were affected by reward condition (*Equation 3a*). For both animals, parameter *k* was significantly increased in large compared to small reward trials ($\chi^2$ likelihood-ratio test of nested models *Equations 3 a,b*: Fle: p < 0.001; Ica: p < 0.001; *Table 1*). This result supports the interpretation that the improvement in performance seen in large reward trials is in part due to an increase in parameter *k*, that is, increased sensitivity of sensory representations to the visual stimulus when reward is large. The estimated distance between the starting point of DV integration and the decision bounds, represented by parameter *B*, was also significantly affected by reward, decreasing in large compared to small reward trials ($\chi^2$ likelihood-ratio test of nested models *Equations 3a,c*: Fle: p = 0.007; Ica: p < 0.001; *Table 1*). Altogether these results indicate that expected reward affects perceptual decision-making both at the sensory representation stage (parameter k) and the integration stage (parameter B). Statistical results are unaffected when data were pooled across all microstimulation sites.

An independent site-by-site analysis confirmed that the effect of microstimulation was significantly decreased in large reward trials compared to small reward trials (*Figure 6*). For each significant PREF microstimulation site, psychometric functions were fit with a pair of cumulative Gaussian functions (*Equation 1a*, 'Materials and methods'), whose means were allowed to differ by microstimulation

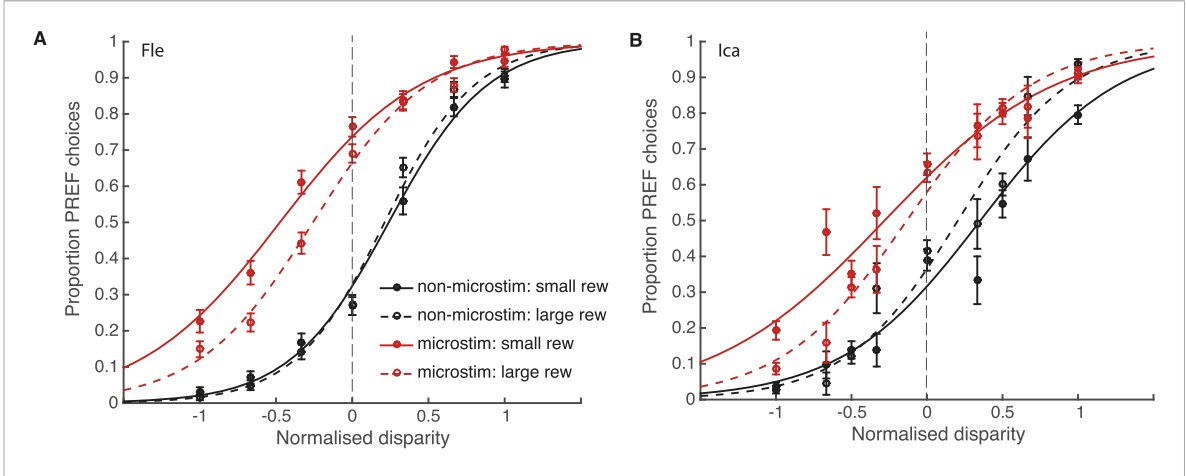

**Figure 5**. Reward modulates the effect of visual cortical microstimulation on perceptual decisions. Trials were pooled across significant PREF microstimulation sites separately for monkeys Fle (**A**) and Ica (**B**). Psychometric functions were fit with the bounded accumulation model (*Equation 3a*). In both cases, the effect of microstimulation (horizontal shift between red and black psychometric functions) was smaller in large reward trials (dashed lines) compared to small reward trials (smooth lines). For both animals, the best-fitting model allowed both parameter *k* (stimulus sensitivity) and parameter *B* (distance to decision bounds) to be affected by reward condition ($\chi^2$ likelihood-ratio tests of nested models *Equations 3b,c*, p < 0.05 in all cases, see main text for details). This suggests that reward can affect sensory representations as well as evidence integration during perceptual decision-making. Error bars show s.e.m.

**Table 1**. Parameter values for bounded accumulation model fit to data combined over significant PREF microstimulation sites

| Animal | Parameter | Estimated value (small reward) | Estimated value (large reward) | % Change in estimate by reward | −logL* for frozen parameter model | −logL* for full model | χ-value of model comparison | p value of χ2-test† |
|--------|-----------|------------------|------------------|------|--------|--------|------|---------|
| Fle | k | 1.74 | 2.41 | +38.3 | 3045.0 | 3034.3 | 21.4 | 0.00002 |
| Ica | k | 1.94 | 3.50 | +80.4 | 2398.9 | 2389.2 | 19.4 | 0.00001 |
| Fle | B | 1.77 | 1.40 | −20.7 | 3037.9 | 3034.3 | 7.2 | 0.0073 |
| Ica | B | 1.29 | 0.79 | −38.4 | 2394.8 | 2389.2 | 11.2 | 0.0008 |

*Negative log likelihoods (−logL) reported for each model fit, that is, with all parameters allowed to vary by reward condition (**Equation 3a**) compared with each parameter frozen (**Equations 3b,c**).

†p values were obtained from a χ2 likelihood-ratio test comparing the fit of the full bounded evidence-accumulation model to a restricted version of the model in which the specified parameter was not allowed to change value for different reward conditions (nested models **Equations 3a,b,c**, 'Materials and methods'; **Figure 5A,B**).

condition but whose standard deviation was constrained to be the same (**Krug et al., 2013**). We illustrate these psychometric functions for example site *fle302*. Here, MU neurons were selective for a CCW cylinder rotation direction (**Figure 6—figure supplement 1A**). The effect of electrical microstimulation was equivalent to adding binocular disparity of −0.023˚ in magnitude to the stimulus dots, biasing choices toward the PREF cylinder direction CCW (**Figure 6—figure supplement 1B**). We then separated trials according to whether available reward was large or small (**Figure 6—figure supplement 1C,D**). Microstimulation always biased the animal's response toward CCW rotation, the PREF rotation direction at this site. However, when reward size was large, the effect of electrical microstimulation was equivalent to addition of −0.019˚ of binocular disparity

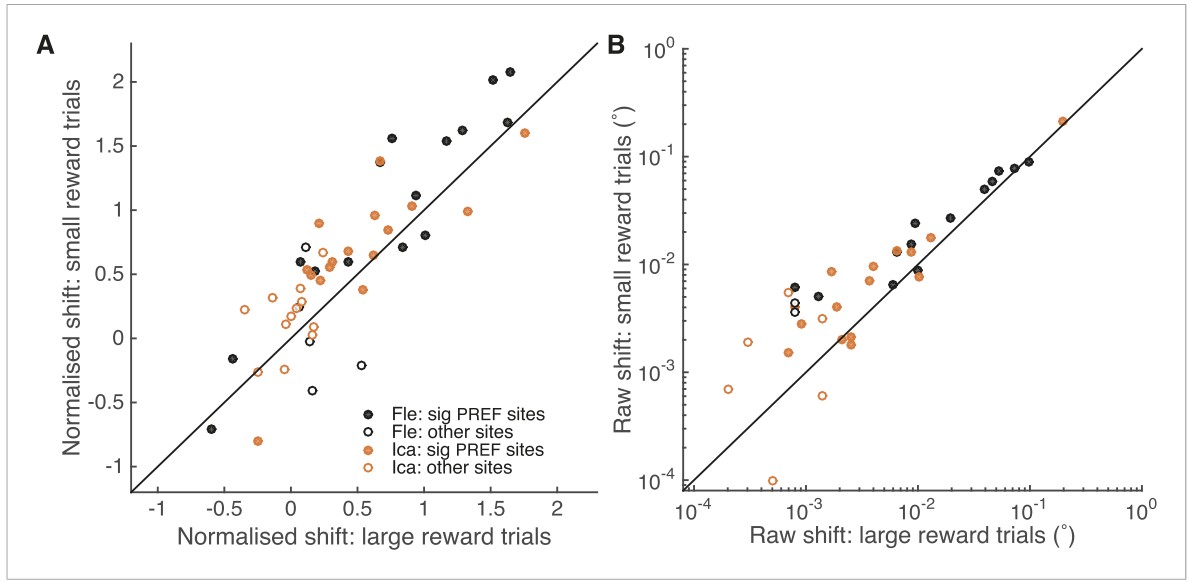

**Figure 6**. A site-by-site analysis confirms that visual cortical microstimulation is less effective at biasing perceptual choices in trials with large expected reward compared to trials with small expected reward. This is significant over both animals for microstimulation shifts normalised by discrimination threshold at each site (**A**; Wilcoxon sign-rank test: p < 0.001) and also for the raw microstimulation shifts (**B**; p < 0.001). This is also significant for each animal separately (see 'Results' for details). Raw microstimulation shifts (**B**) are plotted on a log-scale; please note that the effect of reward is comparable across different microstimulation effect sizes (see **Figure 8**). Black lines indicate the identity relationship.

The following figure supplement is available for figure 6:

**Figure supplement 1**. Effect of reward on visual cortical microstimulation at an example V5/MT site, *fle302*.

(*Figure 6—figure supplement 1C*), representing a reduction of about 30% compared to when reward size was small, where the microstimulation current was equivalent to addition of −0.027˚ (*Figure 6—figure supplement 1D*).

Over all significant PREF microstimulation sites over both animals, we considered both the unit-less normalised microstimulation-induced shift (divided by the cylinder threshold at the site; *Figure 6A*), and the raw shift measured in degrees of visual angle (*Figure 6B*). The normalised microstimulation shift was significantly smaller for large expected reward trials (median = 0.67) compared to small expected reward trials (median = 0.88; Wilcoxon sign-rank test of normalised shift values, two-sided: n = 28, p < 0.001) and also when considered separately for each animal Fle (median large reward trials = 0.94; small reward trials = 1.37; n = 13, p = 0.005) and Ica (median large reward trials = 0.54; small reward trials = 0.68; n = 15, p = 0.022). Similarly, the raw microstimulation shift was significantly smaller for large expected reward trials (median = 0.0064˚) compared to small expected reward trials (median = 0.0093˚; Wilcoxon sign-rank tests of raw shift values: n = 28, p < 0.001) and also when considered separately for each animal Fle (median large reward trials = 0.0099˚; small reward trials = 0.024˚; n = 13, p = 0.013) and Ica (median large reward trials = 0.0025˚; small reward trials = 0.0071˚; n = 15, p = 0.005). Statistical results were unaffected when data were pooled across all stimulation sites.

## Control: visual perturbation to test an adjustment strategy

We tested the possibility that the animals detect some subtle difference between trials with and without electrical microstimulation, and accordingly adjust their decision criterion to bias their choices against the microstimulation effect - and that they might do this more in large reward trials to reduce their error rate. This seems unlikely, as in principle the animals could then have applied a different criterion to all microstimulation trials and thereby eliminated the effect of microstimulation altogether, resulting in far fewer errors overall and thereby improving their acquisition of rewards. Nevertheless, to examine the possibility of such a strategy, we performed a set of psychophysical control experiments with monkey Ica that reproduced the effect of electrical microstimulation using a visual stimulus perturbation. This variation omitted the electrical microstimulation itself, but retained the cognitive cues that might be delivered to the animal during electrical microstimulation.

Using the same experimental parameters as for electrical microstimulation sessions, we inserted an additional binocular disparity signal, 'Δdx', to the cylinder stimulus pseudo-randomly on 50% of trials (see *Salzman et al. 1992*; *Fetsch et al. 2014* for a similar manipulation of visual motion signals in a motion direction discrimination task). In our experiment, the trials with the Δdx signal were overtly flagged by a change in the appearance of the cylinder stimulus from all black dots to all white dots (*Figure 7*). This change in cylinder appearance is highly salient to human observers and we assume it is also salient for the monkey. Importantly, the animal was only rewarded for perceptual choices that would have been correct had the additional disparity signal not been present. This creates a form of 'microstimulation' experiment in which the presence of the extra disparity cue Δdx is clearly signalled by a change in stimulus colour from black to white. The animal was first trained over 10 days and 45,000 trials on this task. Unlike the electrical microstimulation experiments, which contained a mixture of microstimulation biases across different V5/MT sites, favouring sometimes CW and sometimes CCW choices, the added disparity signal Δdx always gave a consistent shift towards CCW perceptual choices. This increased the chance that the animal would engage in a strategy that exploited all available information to reduce its error rate and gain more reward.

Despite the fact that trials with Δdx were clearly signalled, the psychometric function for Δdx trials was horizontally shifted towards the response direction predicted by the additional disparity signal, successfully emulating the horizontal shift of the psychometric function due to V5/MT electrical microstimulation (*Figure 7*). As observed in the electrical microstimulation experiments, there was a significant improvement in performance in trials where a larger reward was available for a correct choice ($\chi^2$ likelihood-ratio test of nested models *Equations 2a,b*: Ica-Δdx: p < 0.001). However, there was no significant interaction between reward condition and Δdx-induced shift in the Δdx psychometric functions ($\chi^2$ likelihood-ratio test of nested models *Equations 2a,c*: Ica-Δdx: p > 0.05; *Figure 7*), in contrast to the results when the same test was applied to the electrical microstimulation datasets ($\chi^2$ likelihood-ratio test of nested models *Equations 2a,c*: Fle: p < 0.001; Ica: p < 0.001). Therefore, there is no evidence to support the view that microstimulation induces a different percept that the animal discounts more effectively when reward is higher, since the animal does not make an adjustment in large reward trials when the presence of additional disparity is clearly signalled.

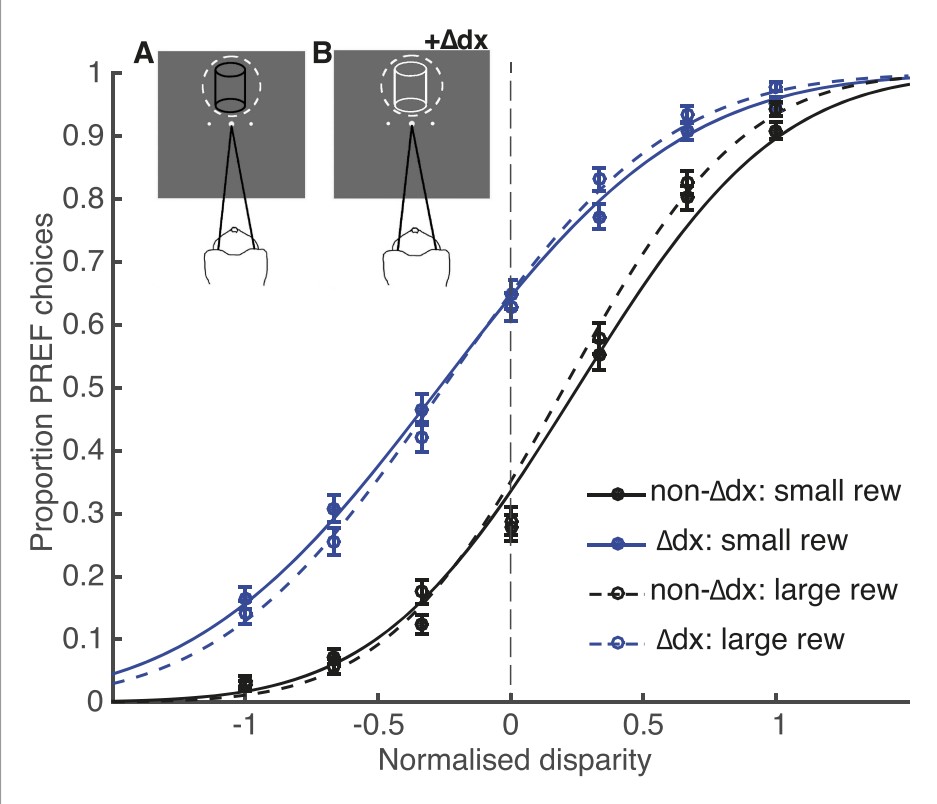

**Figure 7**. Effect of reward on visual cortical microstimulation cannot be explained by a change in the animal's strategy. A control experiment was run with animal Ica, in which microstimulation was simulated by insertion of an additional disparity signal ('Δdx') into the cylinder stimulus. The animal was rewarded only for correct choices with respect to cylinder disparity before insertion of Δdx. Inset panels illustrate the colour change of cylinder dots from all black in trials with no additional disparity added (**A**) to all white dots in Δdx trials (**B**). Performance in the Δdx control was significantly better in large reward compared to small reward trials ($\chi^2$ likelihood-ratio test of nested models **Equations 2a,b**: $p < 0.001$), but there was no significant effect of reward on the shift induced by additional Δdx disparity ($\chi^2$ likelihood-ratio test of nested models **Equations 2a,c**: $p > 0.05$). Therefore, even when trials that contain additional disparity were clearly signalled, the animal did not make an adjustment to counter this additional signal more on large relative to small reward trials. By contrast, under the same model, reward significantly reduced the electrical microstimulation shift for both Fle and Ica (see main text). This suggests that the change in microstimulation shift with reward was not likely to be due to the animals adopting a decision criterion that depended both on reward size and on detecting microstimulation. Error bars show s.e.m.

## Controls: stimulus eccentricity and slower fluctuations in task engagement

We explored further trends in the data to evaluate the potential relationship between the reward modulation of visual cortical microstimulation, stimulus eccentricity, discrimination threshold, and raw overall microstimulation effect at each site. Visual area V5/MT contains a retinotopic map of contralateral visual space (**Albright and Desimone, 1987**) and the receptive field (RF) location of each microstimulated MU varies in eccentricity, that is, distance from the fovea (**Desimone and Ungerleider, 1986**). The cylinder stimulus was matched to the MU RF location at each micro-stimulation site. Both animals' cylinder discrimination thresholds significantly correlated with stimulus eccentricity, worsening as eccentricity increased (**Figure 8A**). The raw overall microstimulation effect also significantly correlated with the cylinder discrimination threshold at each site (**Figure 8B**). The reward modulation of the raw microstimulation shift effect was significantly correlated with stimulus eccentricity and with the raw overall microstimulation effect size for both animals considered together, and for one animal considered alone (Ica), such that the biggest raw effects of reward on microstimulation occurred for sites with largest eccentricity (where thresholds were high) and the

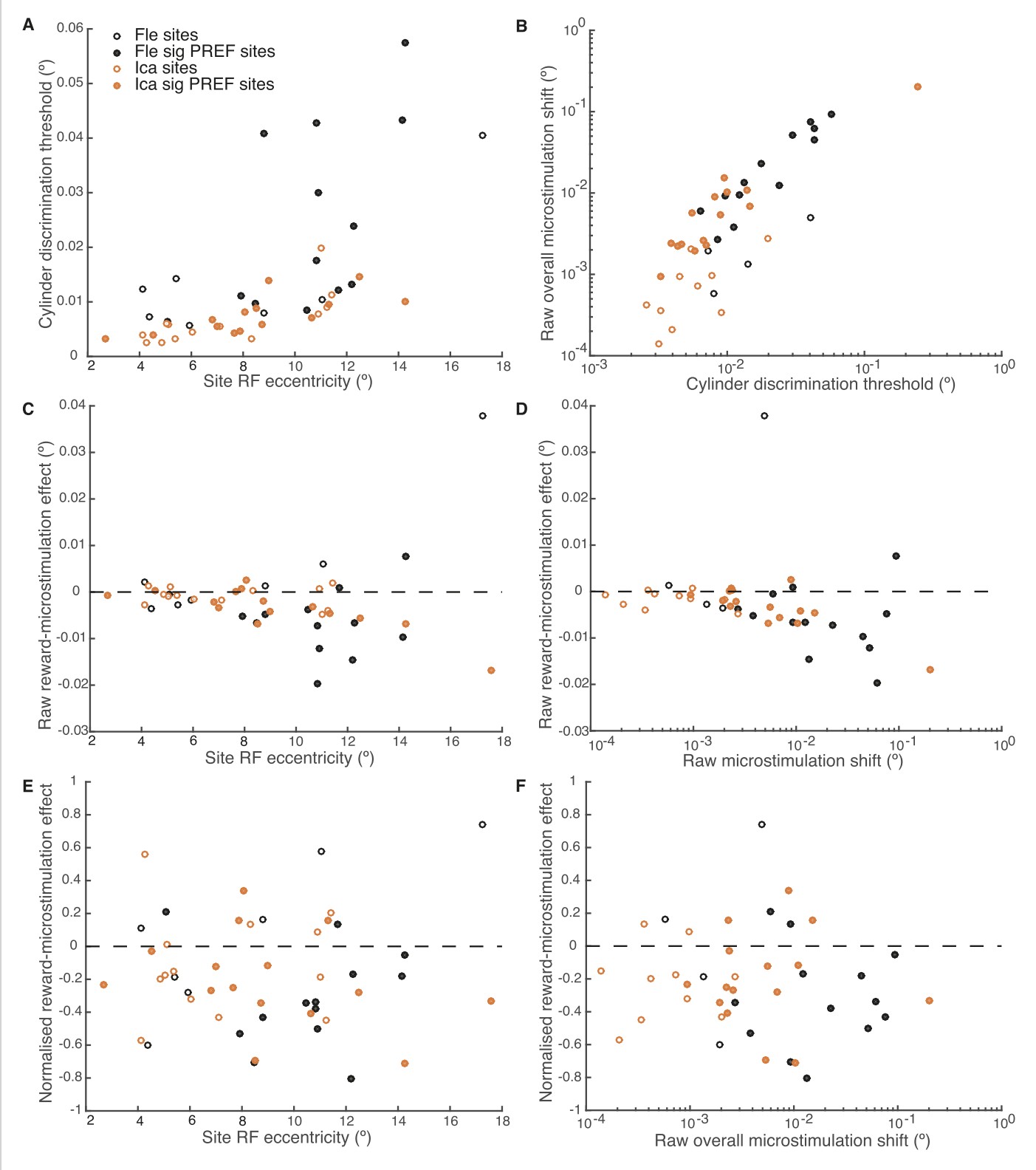

**Figure 8**. Effect of reward on visual cortical microstimulation is constant over stimulus eccentricity and over raw overall microstimulation effect size. (**A**) Cylinder discrimination threshold (s.d. of the cumulative Gaussian fitted to the psychometric function) significantly correlates with stimulus eccentricity, which is determined by the MU receptive field (RF) location at the microstimulation site (Fle: p = 0.001; Ica: p = 0.001). (**B**) Raw overall microstimulation effect at each site is significantly correlated with discrimination threshold (Ica: p < 0.001; Fle: p < 0.001). (**C**) Effect of reward on the raw microstimulation shift (i.e. the raw microstimulation shift in large reward trials minus the raw shift in small reward trials) significantly correlates with stimulus eccentricity for

*Figure 8. continued on next page*

*Figure 8. Continued*

both animals considered together (p = 0.018) and also for Ica alone (p < 0.001). (**D**) Effect of reward on the raw microstimulation shift significantly correlates with the raw overall microstimulation effect for both animals considered together (p = 0.028) and also for Ica alone (p < 0.001). (**E**) When the effect of reward is considered over microstimulation shifts normalised by discrimination threshold, the correlation with stimulus eccentricity disappears (p > 0.05 in all cases). (**F**) When the effect of reward is normalised by discrimination threshold at each site, the correlation with raw overall microstimulation effect disappears (p > 0.05 in all cases). (**E**, **F**) show that the reward effect is constant when expressed as a fraction of discrimination threshold. All correlation tests use Pearson's product–moment correlation coefficient.

The following figure supplements are available for figure 8:

**Figure supplement 1**. Fluctuations in performance do not explain effect of reward on visual cortical microstimulation.

**Figure supplement 2**. Waning of microstimulation does not explain reward effect.

**Figure supplement 3**. Evaluation of model fits with quantile–quantile comparisons.

biggest raw overall microstimulation effect (*Figure 8C,D*). However, these correlations disappeared for reward modulation of the microstimulation shift effect when normalised by cylinder discrimination threshold at each site (*Figure 8E,F*). This shows that the reward effect is constant when microstimulation is expressed as a fraction of the discrimination threshold.

In these experiments, available reward size for a correct choice depended upon animals' performance in the preceding trials, increasing gradually when animals performed correctly over consecutive trials (*Figure 2*). Sequences of correct trials are more likely when the animal is more engaged in the task, perhaps due to attentional or motivational states. In an alternative interpretation, slower fluctuations in task engagement, rather than expected reward size, are the cause of the observed effects. Variations in task engagement would be expected to occur at least an order of magnitude slower (over the space of around 30 trials) than differential valuations of upcoming reward (which occur over the space of two or three trials; *Figure 2*). We therefore address this concern with a time-series analysis of task engagement.

For the experimental session at each microstimulation site for each animal, a smoothed performance average was calculated using a sliding time window of size 30 trials (*Figure 8—figure supplement 1A*). The extent of performance fluctuation over the experimental session was estimated as the area between the smooth performance curve and average (normalised by total number of trials). If fluctuations in task engagement underlie the observed association between reward and microstimulation shift, there should be a correlation between the extent of performance fluctuation and size of reward effect. However, there was no correlation for either animal considered separately or together (p > 0.05 in all cases; *Figure 8—figure supplement 1B*). Furthermore, the effect of expected reward on microstimulation shift remained when trials were separated according to whether they occur in 'good' or 'bad' epochs of task engagement ($\chi^2$ likelihood-ratio test of nested models *Equations 2a,c*, data pooled across sites and animals: good epochs: p < 0.001; bad epochs: p < 0.001; *Figure 8—figure supplement 1C,D*). Therefore, the modulation of cortical microstimulation by available reward size does not seem to be associated with or dependent upon slower fluctuations in task engagement.

In further controls, we found that the effect of reward could not be explained by the waning of microstimulation effect over time during an experimental session, combined with a propensity to have a greater proportion of large reward trials at the end of a microstimulation session (*Figure 8—figure supplement 2*). Also, there were no significant differences in proportion of microstimulated trials or mean absolute disparity between reward conditions (*Supplementary file 1*).

## Discussion

Performance-contingent reward size modulated the effect of electrical microstimulation of visual area V5/MT on visual perceptual decisions. The size of the perceptual shift induced by electrical microstimulation was smaller when the available reward for a correct choice was larger. These results suggest that reward signals interact differently with visually evoked and artificially introduced sensory information. A simplified bounded accumulation model of perceptual decision-making best explains the data if expected reward is allowed to affect the representation of sensory evidence. This effect of reward is in addition to an effect upon the integration of that evidence during the formation of the

perceptual decision. If reward were to affect only the integration stage, it should influence behavioural responses to neural signals in the same way regardless of whether those signals have been evoked by visual input or artificially by microstimulation (*Figure 4*). Experimentally, however, we found the converse (*Figure 5*). We propose that, on this view, reward interacts with the sensory/perceptual signals in cortical representations of the visual stimulus during visual perceptual decision-making, in addition to affecting the integration stage (see *Figure 1*).

In the fitting of the bounded evidence-accumulation model, we found that when available reward was large there was a decrease in the distance between the starting point of the DV and the decision bounds. Under conditions where one choice is associated with a larger expected reward than the other, previous neurophysiological, human neuroimaging and computational studies have shown a decreased distance between the DV starting position and the decision bound for larger reward, which is beneficial because this bias results in an overall greater accumulation of reward (*Feng et al., 2009*; *Rorie et al., 2010*; *Summerfield and Koechlin, 2010*; *Gao et al., 2011*; *Mulder et al., 2012*). When *both* choices are associated with larger rewards, as in the present study, the distance to *both* decision bounds might be expected to decrease, as we, and others (*Rorie et al., 2010*), have found. On its own, this effect would impair performance accuracy when available reward is large because the closer proximity of the DV to the bounds would result in a greater effect of accumulated noise on the final perceptual choice (*Bogacz et al., 2006*, *2007*; *Rorie et al., 2010*). However, the reciprocal increase in sensitivity to the visual stimulus, discussed above, balances the decrease in accuracy resulting from the shorter distance-to-bound, so that overall performance in fact improves when available reward is larger, as we observe.

Previous studies found that information about reward was reflected only at the integration stage of visual perceptual decision-making, represented in sensorimotor structures (*Feng et al., 2009*; *Rorie et al., 2010*; *Summerfield and Koechlin, 2010*; *Gao et al., 2011*; *Mulder et al., 2012*). However, experiments employing other types of task, such as a cued curve-tracing saccade task (*Stanisor et al., 2013*), somatosensory discrimination (*Pleger et al., 2008*), tasks in which a subject must select between two presented visual stimuli (*Serences, 2008*; *Serences and Saproo, 2010*), or a fixation task where a reward is delivered with or without a visual cue (*Arsenault et al., 2013*), found that activity in sensory cortex can also be modulated by reward. Except for *Stanisor et al. (2013)*, these studies used BOLD-fMRI, an indirect measure of neural activity that averages the responses of many neurons excited or inhibited by the sensory stimulus (*Logothetis et al., 2001*; *Logothetis and Wandell, 2004*). Our results extend these previous findings by indicating that expected reward also interacts with sensory representations about the visual stimulus during visual discrimination itself.

The source of the reward signal affecting sensory cortex may reflect an influence from prefrontal cortex, anterior cingulate cortex, or amygdala, known to represent reward and value predictions during decision-making (*Rushworth and Behrens, 2008*; *Bermudez and Schultz, 2010*; *Levy and Glimcher, 2012*), or neuromodulation by serotonergic neurons originating from the dorsal raphe nucleus, which also signal reward (*Cohen et al., 2015*). Another significant source of reward signals in the brain are dopaminergic reward-predicting neurons, which may also indirectly modulate or enhance sensory cortical activity (for review, see *Schultz 2002*).

In the present study, the size of available reward was expected to affect animals' motivation and indeed they demonstrate better performance in large reward trials. In this way, our study differs from other studies in which prior probability, not reward value, of each stimulus is cued in each trial, which may bias perceptual decisions to obtain larger overall rewards without necessarily affecting motivation or performance (for example, *Rao et al. 2012*).

## Controls for animal strategy and performance fluctuation

Our results could not be explained by an adaptive change in the animals' strategy based upon recognizing the presence of microstimulation. In a control experiment, in which an additional visual disparity (Δdx) signal that mimicked the microstimulation effect was clearly cued by a change in stimulus colour, the effect of Δdx on choices was not affected by expected reward size, even though overall task performance was better in large reward trials as seen in the electrical microstimulation sessions. This indicates that the animal did not apply a strategy to compensate for the Δdx signal specifically on large reward trials, even when trials that contained additional visual disparity were clearly identifiable. This experiment controls for the situation in which microstimulation produces a

different percept that the animal is able to discount more effectively when available reward is higher. However, this psychophysical experiment cannot simulate possible noise or other aberrance in inputs associated with electrically evoked sensory neuronal activity, which, for example, might be weighed less compared to more 'normal', visually evoked inputs; weights that could change based on expected reward. We discuss this issue further in the next section.

In the present study, reward depended on animals' performance in the preceding trials, increasing gradually when animals performed correctly over consecutive trials, which is more likely to happen when the animal is more engaged in the visual task. Therefore, instead of expected reward size, fluctuations in task engagement (for example due to attentional or motivational states) could result in epochs of good and bad performance, which are inevitably associated with more large reward or small reward trials respectively. These fluctuations could therefore potentially underlie the observed association between large reward trials and better performance, and the observed effects on the microstimulation shift. However, a time-series analysis of performance over the course of the microstimulation experimental sessions showed that there was no correlation between the performance fluctuation and the differential effect of microstimulation by reward. Furthermore, when trials in each microstimulation session were separated according to whether they occurred in a 'good epoch' (a window of better-than-average performance) or 'bad epoch' (poorer-than-average performance), the modulation of microstimulation shift by reward size remained within both types of epoch. These analyses, which monitor closely engagement with the task over time, provide evidence against the hypothesis that the differential microstimulation effect can be accounted for purely by fluctuations in task engagement. Future experiments can directly rule out this possibility by varying reward size independently of performance, for example by using a block design or trial-by-trial cueing of the available reward size.

## Limitations of a simplified bounded evidence-accumulation model

In our interpretation of the results, we assume that electrically evoked signals in visual area V5/MT are thereafter integrated into the perceptual decision-making process in a manner identical to visually evoked evidence (*Figure 5*). This interpretation is supported by the behavioural shift induced by electrical microstimulation in perceptual decisions. However, the circuit mechanisms of the effect of electrical microstimulation are incompletely understood, and microstimulation has been reported to have unanticipated inhibitory effects on downstream cortical targets (*Logothetis et al., 2010*; *Sultan et al., 2011*). Such observations have generally been made in anaesthetised animals, without the concurrent presentation of a visual stimulus expected to activate the same brain areas as microstimulation, and at stimulation currents orders of magnitude higher than used in this study (250–1000 μA, compared to our 20 μA). Therefore, it is difficult to interpret these effects in the context of our experimental protocols.

A remaining possibility is that odd or aberrant inputs from electrically evoked sensory neurons might be weighed less by decision-making areas compared with more 'normal' that is, visually evoked inputs, and that these weights could change based upon expected reward. Our $\Delta dx$ experiment cannot control for this case because we added the additional stimulus disparity signal via visual stimulation that by this argument cannot mimic the artificially induced microstimulation signal. However, we argue that there is currently little evidence to suggest that the brain applies differential weights to electrically and visually evoked sensory signals in this way. It is important to note that microstimulation affects behaviour in both large and small reward trials, suggesting that it is not an aberrant signal but a clear signal that affects perceptual choices in all cases. When combined with relevant visual stimulation, electrical microstimulation in many visual cortical areas significantly and reliably affects behaviour in a variety of visual perceptual tasks (*Cicmil and Krug, 2015*). If the microstimulation signal could affect final motor output via a neural path independent from visually evoked signals, it would be expected that microstimulation would be detectable and/or processed by the brain in some qualitatively different way. But a recent study that modelled in detail the relationship between microstimulation, perceptual choices and decision confidence found that their data did not support the view that microstimulation induces a change in neural activity that is processed qualitatively differently from visually evoked activity (*Fetsch et al., 2014*), and our $\Delta dx$ control shows that the animal cannot use an obvious difference in the stimulus percept to discard some of the visual information provided.

Electrophysiological recordings from stimulus-relevant neurons during visual perceptual decision-making are necessary to directly show whether expected reward modulates the neuronal firing rates and tuning functions at a specific visual cortical site in the manner predicted by an increase in stimulus sensitivity under large reward. Yet, the unique contribution of a microstimulation intervention to this question is to show that reward modulations at a level prior to the integration stage have a *causal* effect on the perceptual decision, rather than representing a top-down echo of decision-formation that might not be causally relevant to the perceptual choice itself.

The form of the bounded accumulation model considered here is a simplified case of the race model, in which evidence for each choice is integrated separately by two decision variables (*Usher and McClelland, 2001*). We used the assumption that the accumulator inputs are perfectly negatively correlated, which corresponds to the action of one accumulator only (*Figure 5A,B*) (*Palmer et al., 2005*; *Ratcliff and McKoon, 2008*). To avoid unnecessary assumptions and over-fitting, we chose the simplest model framework that could explain our findings. The microstimulation signal was modelled as additional sensory evidence towards the cylinder rotation direction preferences of the micro-stimulated neurons. The null choice bias, whereby animals apply an overall bias towards the non-preferred direction of the stimulated neurons in order to match overall reward proportions for different choices (*Salzman et al., 1992*), was modelled as an offset to the overall psychometric function (*Equation 3a*, 'Materials and methods'). The parameters of the bounded accumulation model are commonly used to explain both choice and reaction time. However, bounded integration underlies perceptual decisions even when stimulus viewing duration is dictated by the environment (*Kiani et al., 2008*), and the model can be fit usefully in these circumstances (*Fetsch et al., 2014*). Although we employed a fixed viewing duration task, differential effects of reward on parameter *k,* coding for stimulus scaling sensitivity, and parameter *B*, coding for distance-to-bound, could be differentiated in our model due to the insertion of the microstimulation signal in visual area V5/MT (*Figure 4*).

The lack of reaction-time measurements in the present experiment leaves open the possibility that changes to parameter *B* may also arise from an overall increase in the gain of accumulation of sensory evidence. The consensus from electrophysiological studies (*Rorie et al., 2010*), human neuroimaging (*Summerfield and Koechlin, 2010*; *Mulder et al., 2012*), and psychophysical studies (*Diederich and Busemeyer, 2006*; *Diederich, 2008*) is that reward effects on the DV accumulation process occur through changes to the distance-to-bound; we therefore follow this interpretation. A microstimulation-reaction time task, or detailed measurement of neuronal firing rates in sensorimotor structures such as LIP that represent the integration stage, would be necessary to resolve this issue. Regardless, the specific interpretation of parameter *B*'s effects does not affect the main conclusions of this study.

## Relationship between reward and attention

We have argued that the most parsimonious explanation for our observations is that larger available reward increases the sensitivity of visually evoked sensory representations about the visual stimulus. This bears strong similarity to the effect on neuronal responses when attention is allocated to a specific spatial location or to a PREF stimulus feature (*Treue and Maunsell, 1996*, *1999*; *McAdams and Maunsell, 2000*; *Corbetta and Shulman, 2002*; *Saenz et al., 2002*). Indeed, manipulations of reward and attention are often difficult to separate (*Maunsell, 2004*). However, these effects cannot trivially explain our findings.

With regard to feature attention, our trial-by-trial reward schedule was balanced so that both perceptual interpretations of the cylinder were equally rewarded, if correctly identified. Feature attention was therefore not directed in favour of one perceptual interpretation of cylinder rotation direction. Our experimental task always directed spatial attention to the location of the visual stimulus, because the animal responded to the visual stimulus in every trial to obtain a reward, whether small or large. Thus, results from spatial attention studies, which measure differences in neuronal activity when attention is directed into and away from their RF (*Treue and Maunsell, 1996*, *1999*; *McAdams and Maunsell, 2000*), cannot directly be applied to our results. The definition of spatial attention could be extended to encompass gradations of attention applied to a particular spatial location according to expected reward size, which may be manipulated by experiments such as ours. In that case, the present study would be the first to provide data demonstrating such an effect during perceptual decision-making, and to integrate such modulations of sensory representations coherently with reward manipulation and decision model theory (*Smith and Ratcliff, 2009*).

In summary, we found that the effect of V5/MT electrical microstimulation on perceptual decisions was less effective when available reward was larger. In the context of a bounded evidence-accumulation model, this suggests that reward modulates sensory representations during perceptual decision-making, in addition to altering the integration of sensory evidence into the DV in sensorimotor structures (*Gold and Shadlen, 2007*). Further electrophysiological research should record in area LIP to show directly how the electrically introduced signal in V5/MT is integrated into the perceptual decision under different reward states, and should record in visual area V5/MT to show how reward modulates the signals of sensory evidence in visual cortex during perceptual decisions.

## Materials and methods

### Animals

Microstimulation experiments were conducted in one cortical hemisphere in two adult male rhesus macaque monkeys (*Macaca mulatta*). Prior to the experiments, each monkey was surgically implanted under general anaesthesia with a head-holding device and a recording chamber placed over a craniotomy above the occipital lobe. For monkey Fle, scleral magnetic search coils were implanted in both eyes under general anaesthesia, to monitor eye position and vergence. Monkey Ica had one scleral magnetic search coil only. Monkeys were trained to fixate on a binocularly presented marker and to perform the visual task that required the animals to discriminate the motion and binocular disparity information in a rotating SFM cylinder stimulus (*Figure 2*).

### Locations and ethics statements

Experiments were conducted at two locations: monkey Fle was tested at Oxford University, UK; monkey Ica was tested at the National Institutes of Health, Bethesda, USA (henceforth referred to as Oxford and NIH, respectively). Except where specifically described, the protocols were identical in the two locations.

At Oxford, all procedures complied with United Kingdom Home Office regulations on animal experimentation. At NIH, all procedures complied with US Public Health Service policy on the humane care and use of laboratory animals, and all protocols were approved by the National Eye Institute Animal Care and Use Committee.

### Visual stimuli

The visual stimulus for the discrimination task was a rotating SFM cylinder made up of two transparent surfaces of random dots moving in opposite directions (random dot kinematogram) (*Treue et al., 1991*) (*Video 1*). The dots had a sinusoidal velocity profile as would be expected in an orthographic projection of a three-dimensional rotating cylinder. When the two dot surfaces are presented at the same binocular disparity, the cylinder rotation direction is bistable and over time will undergo spontaneous fluctuations in the perceived direction of rotation (*Wallach and O'Connell, 1953*; *Ullman, 1979*). Adding opposite horizontal disparity separates the front and back walls of the cylinder in depth, thereby disambiguating the direction of rotation. The disparity of each dot was scaled sinusoidally to match the velocity profile, with the maximal disparity difference between the two dot surfaces at the central axis of the cylinder. Equal but opposite disparities were added to the front and back surfaces of the cylinder, so that its principal axis remained in the plane of fixation. Below, we use the term 'disparity' to refer to this maximal difference in disparity between dots moving in opposite directions. We used the following convention: when the front wall of dots moved to the left with respect to cylinder orientation axis, this corresponded to a CW rotating cylinder as viewed from above; when the front wall moved to the right, this corresponded to a CCW rotating cylinder. To allow pooling across microstimulation sites, quantitative behavioural analysis was carried out with reference to PREF choices. For the pooled data, positive disparities refer to the stimulus rotation direction for which the site was selective.

Cylinder stimuli consisted of 50% black dots and 50% white dots presented at maximum contrast (99% of full contrast) on a mid-grey background at a dot density of 25%. Half the dots (randomly selected) moved in one direction and the remaining dots moved in the opposite direction. When a dot reached the edge of the cylinder, it reversed direction.

In Oxford, stimuli were displayed binocularly using a Wheatstone stereoscope configuration with two monitors (Eizo FlexScan 78, Eizo, Bracknell, UK). A pair of mirrors positioned in front of the monkey reflected the images from each of the two monitor screens separately into the left and right eyes. Frame rate was 72 Hz. Monitor screens were positioned 84 cm from the monkey, and covered approximately 21° × 17° of the visual field. Mean luminance was 42 cd/m² and dot diameter was 0.2°. The dot refresh rate was 2%, that is, on each video frame, 2% of the dots were relocated to a randomly chosen location on the cylinder surface, so that dots had a mean lifetime of 615 ms in each location.

At NIH, stimuli were displayed binocularly using two DLP projectors (Projection Design evo2sx+, projectiondesign®/Barco, Inc., Xenia, OH) with polarizing filters. The image was projected onto a polarisation-preserving screen (Filmscreen 150, Stewart Filmscreen, Amelia, OH). Frame rate was 60 Hz. The screen was 112 cm away from the monkey, and covered approximately 41° × 32°. Mean luminance of the display viewed through polarized filters was 17.5 cd/m² and dot diameter was 0.18°. Dot refresh rate was 1%.

## Psychophysical task

The psychophysical task was a two-alternative forced choice discrimination of the direction of rotation of the SFM cylinder. Stimulus disparities were matched to the psychophysical threshold of the animal in order to obtain a full psychometric function. To initiate a trial, the animal had to acquire the fixation point. The visual stimulus was then presented for 2000 ms (*Figure 2*). At stimulus offset, the fixation point also disappeared and two choice targets corresponding to the two possible directions of cylinder rotation appeared. In the Oxford, targets were located to the left and right side of the fixation point. In the NIH set-up, targets were also located opposite one another on either side of the fixation point but on an axis normal to the cylinder orientation. The animals indicated their perceptual decision by making a saccadic eye-movement to one of the two targets. A saccade to the correct target resulted in a fluid reward. If the choice was incorrect, the animal received no reward and there was a short time-out before the start of the next trial. For ambiguous cylinders (zero-disparity trials), 50% of the trials were rewarded at random. If the animal broke fixation during stimulus presentation, no reward was given.

## Reward schedule

Animals worked on the task to gain fluid rewards to meet their daily requirements. Reward size available for a correct choice on each trial depended upon the number of immediately preceding consecutive correct responses that the animal had made, increasing in two steps up to a maximum (*Figure 1*). Choices in zero-disparity trials were rewarded 50% of the time at random, and for the purposes of calculating the reward sequence they were discounted. For monkey Ica, reward size was 0.08 ml on the first and second consecutive correct choices after an error, 0.12 ml for the third consecutive correct choice, and 0.2 ml on the fourth and all subsequent consecutive correct trials. For monkey Fle, reward size was 1/3 of maximum for the first correct choice, 2/3 of maximum for the second, and reached maximum size (usually 0.18 ml) for the third and all subsequent consecutive correct choices. Thus reward size increased more quickly for Fle than Ica as a function of consecutive correct responses. For both animals, we categorized trials into two conditions: maximal (large) reward size and sub-maximal (small) reward size, where for both animals the average size of the two sub-maximal reward sizes was half that of the maximal reward size. Animals were familiar with their respective reward schedule because it was used throughout all training and recording sessions with the discrimination task; it was not introduced only for the microstimulation experiments.

## V5/MT multi-unit recording

Recording from MU neuronal clusters was carried out to characterize and select the cortical microstimulation sites. Single tungsten microelectrodes coated with polyimide tubing were used (0.1–0.3 MΩ impedance at 1 kHz; MicroProbe, Inc., Gaithersburg, MD). A hydraulic microdrive mounted on the recording cylinder advanced the electrode through a guide tube. Electrical signals from the electrode were filtered, amplified and displayed through visual and audio monitors, and stored to computer disk. Online classification of activity and pre-processing steps were done with

the DataWave Discovery system (DataWave Technologies, Loveland, CO) at Oxford and Spike 2 (Cambridge Electronic Design Ltd., Cambridge, UK) data acquisition program at NIH.

Area V5/MT was approached posteriorly through the recording chamber in incremental steps of about 100 μm. Area V5/MT was identified by established physiological criteria: (1) the approach through the grey/white matter pattern comprising the striate cortex, lunate sulcus and the posterior bank of the superior temporal sulcus (STS) (*Zeki, 1974*); (2) by neurons' characteristic direction selectivity and binocular disparity selectivity and its clustering in V5/MT (*Van Essen et al., 1981*; *Maunsell and Van Essen, 1983*; *Albright and Desimone, 1987*; *Bradley et al., 1995*, *1998*; *DeAngelis et al., 1998*; *DeAngelis and Newsome, 1999*); (3) the known relationship between the size and eccentricity of V5/MT receptive fields (*Desimone and Ungerleider, 1986*); and (4) from penetration to penetration, the relationship between RF positions and known topography of V5/MT (*Albright and Desimone, 1987*).

## Selection and characterisation of stimulation sites

Monkeys maintained fixation on a central fixation point while direction selectivity was established for an MU cluster at a particular electrode location, using kinetic dots moving coherently in different directions. The direction evoking the greatest response on average was assigned to be the PREF motion direction for the MU cluster. The RF boundaries were then mapped using a patch of dots moving in the PREF motion direction. After quantitative confirmation of direction tuning within these boundaries, the binocular disparity preference of the MU cluster was measured by presenting a plane of moving dots in the PREF direction at different binocular disparities around the fixation plane. Only MU sites with discernible motion and disparity tuning were further explored. The directions of motion of the two transparent planes of dots that made up the cylinder were aligned with the PREF and the opposite (NULL) directions of the MU cluster and the MU cylinder preference was calculated online over a range of disparities.

When the direction- and cylinder disparity-preference of MU cluster activity were found to remain constant over at least 300 μm of cortex, the electrode was retracted to the middle of that stretch. At the proposed stimulation site, the MU RF size, direction selectivity and cylinder preference were quantitatively re-measured. The cylinder stimulus position, size, orientation, and dot speed were carefully matched to the preferences of the MU at the stimulation site. The monkey first performed 50 to 140 trials of the psychophysical task without microstimulation, prior to the microstimulation experiment, in order to allow the selection of a range of stimulus disparities near threshold at that site.

The total number of cortical sites at which microstimulation was applied was 20 in monkey Fle and 28 in monkey Ica. This number of biological replicates (different stimulation sites) was chosen to be as close as possible to the number of microstimulation sites used in previous studies of this cortical area (V5/MT; in particular, *DeAngelis et al. 1998*). Stimulation sites were selected freshly on each day of recording and the choice of sites for stimulation was independently determined on each occasion of performing the experiment, by reference to the selection criteria presented above.

## Experimental procedure for microstimulation

Electrical microstimulation was applied during the 2000 ms of stimulus presentation on half of the trials, pseudo-randomly interleaved with non-microstimulated trials. Stimulation consisted of 20 μA biphasic pulses of 200 μs cathodal stimulation followed by 200 μs anodal stimulation, delivered at 200 Hz. Only cortical microstimulation sites with at least 10 microstimulation trials and 10 non-stimulation trials at each of at least 5 different stimulus disparities were included in subsequent analyses. All included sites showed significant tuning to disparity in the cylinder stimulus (one-way ANOVA, $p < 0.05$).

## Site-by-site quantification of microstimulation effect

For each microstimulation site, we plotted the proportion of choices made by the monkey towards the PREF rotation direction, at each stimulus disparity, separately for microstimulated and non-microstimulated trials. Behavioural data were fitted for each site with a two-mean cumulative Gaussian distribution:

$$\mathrm{P_{PREF}}(C) = 1\big/2\Big(1 + erf\Big(\big(C - (\mu_0 + \beta\mu_1)\big)\big/\sigma\sqrt{2}\Big)\Big), \tag{1a}$$

where $C$ is the cylinder disparity (positive in PREF direction); $\mu_i$ is the mean of the distribution; $\sigma$ is the s.d.; $erf$ is the error function; $\beta = 1$ for microstimulated trials and $\beta = 0$ for non-microstimulated trials. $P_{PREF}$ corresponds to the probability of making a decision in the PREF direction of the microstimulated site. To test whether microstimulation significantly shifted the psychometric function, the two-mean model was compared to a model in which mean was not allowed to vary by microstimulation (*Krug et al., 2013*):

$$P_{PREF}(C) = 1\Big/2\Big(1 + erf\big((C - \mu_0)\big/\sigma\sqrt{2}\big)\Big). \tag{1b}$$

A $\chi^2$ likelihood-ratio test was used to ascertain whether the full model fitted the data significantly better than the nested model ($p < 0.05$; 1 degree of freedom as models differed by 1 parameter). Maximum likelihood estimate model fitting was performed with MATLAB's *fminsearch* function and was repeated multiple times from a wide range of starting parameter values, in order that results would not be driven by local minima.

The microstimulation effect is quantified by taking the estimated value of parameter $\mu_1$; this is equivalent to a horizontal shift in the psychometric curve. The size of the electrical microstimulation effect can thus be expressed in terms of the size of binocular disparity that would need to be added to the stimulus to shift perceptual behaviour to the same extent (*Krug et al., 2013*).

### Quantification of interaction between reward, task performance and microstimulation

A cumulative Gaussian function was used to evaluate the interaction between reward, task performance and microstimulation for psychometric data pooled across multiple microstimulation sites. Prior to pooling, we normalized stimulus disparity values at each site by dividing them by the maximum disparity value of the site. This ensured that the maximum normalized disparity was 1 and minimum was −1, with the other disparity values lying in between. In the full model both the mean $\mu$ and standard deviation (s.d.) $\sigma$ are allowed to vary by microstimulation condition and reward condition,

$$P_{PREF}(C) = 1\Big/2\Big(1 + erf\big((C - (\mu_0 + \beta\mu_1 + \alpha\mu_2 + \alpha\beta\mu_3))/(\sigma_0 + \alpha\sigma_1 + \beta\sigma_2)\sqrt{2}\big)\Big), \tag{2a}$$

where $C$ is the cylinder disparity (positive in PREF direction); $\mu_i$ represents the mean; $\sigma_i$ represents s.d., $\alpha = 1$ in trials were available reward is large, otherwise $\alpha = 0$; $\beta = 1$ in trials where additional stimulation is introduced, otherwise $\beta = 0$; $erf$ is the error function. $P_{PREF}$ corresponds to the probability of making a decision in the PREF direction of the additional stimulation introduced. To test whether reward affected task performance, this full model was compared with a nested model in which s.d. $\sigma$ (a measure of performance threshold) was not allowed to vary by reward condition:

$$P_{PREF}(C) = 1\Big/2\Big(1 + erf\big((C - (\mu_0 + \beta\mu_1 + \alpha\mu_2 + \alpha\beta\mu_3))\big/(\sigma_0 + \beta\sigma_2)\sqrt{2}\big)\Big). \tag{2b}$$

A $\chi^2$ likelihood-ratio test was used to ascertain whether the full model fitted the data significantly better than the nested model ($p < 0.05$; 1 degree of freedom as models differed by 1 parameter). Similarly, the effect of reward on microstimulation was evaluated with a $\chi^2$ likelihood-ratio test to compare the full model with a nested model in which the change in mean by microstimulation, represented by $\beta\mu_i$, was not allowed to vary by reward condition:

$$P_{PREF}(C) = 1\Big/2\Big(1 + erf\big((C - (\mu_0 + \beta\mu_1 + \alpha\mu_2))\big/(\sigma_0 + \alpha\sigma_1 + \beta\sigma_2)\sqrt{2}\big)\Big). \tag{2c}$$

Model fitting was performed as described in the previous section. We report uncorrected p values; however, all statistically significant results survive Bonferroni correction for multiple comparisons across the two animals (where relevant).

### Decision model analysis: bounded evidence accumulation

A logistic regression model representing the effect of cylinder stimulus disparity, microstimulation and trial reward condition on the psychometric functions was derived from the simple, one-dimensional drift-diffusion model of perceptual decision-making (*Palmer et al., 2005*; *Gold and Shadlen, 2007*)

(see also *Figure 1*). The full bounded evidence-accumulation model explains perceptual choices in the PREF direction ($P_{PREF}$) according to the following equation:

$$P_{PREF}(C) = 1 \, / \left(1 \, + \, \hat{e}(-((B_0 \, + \, \alpha B_1)((k_0 + \alpha k_1 + \beta k_2)C + \beta) + x_0 + \alpha x_1))\right), \quad (3a)$$

where $C$ is the (normalized) cylinder disparity (positive in PREF direction); $\alpha = 1$ for large reward trials and $\alpha = 0$ for small reward trials; $\beta = 1$ for microstimulated trials and $\beta = 0$ for non-microstimulated trials. The relation between stimulus strength C and the aspect of DV drift that is driven by visually evoked sensory evidence is represented by parameter $k_i$ (see also *Figure 4A,B*). The distance to the decision bounds from the starting point of the drift-diffusion process and the overall gain of the drift rate are represented by parameter $B_i$. Behavioural *null bias* in non-stimulated trials (*Salzman et al., 1990*, *1992*) is represented by parameter $x_i$. To test whether reward affected parameter $k_i$, this full model was compared with a nested model in which $k_i$ was not allowed to vary by reward condition:

$$P_{PREF}(C) = 1 \, / \left(1 \, + \, \hat{e}(-((B_0 \, + \, \alpha B_1)((k_0 + \beta k_2)C + \beta) + x_0 + \alpha x_1))\right). \quad (3b)$$

A $\chi^2$ likelihood-ratio test was used to ascertain whether the full model fitted the data significantly better than the nested model ($p < 0.05$; 1 degree of freedom as models differed by 1 parameter). Similarly, the effect of reward on parameter $B_i$ was evaluated with a $\chi^2$ likelihood-ratio test to compare the full model with a nested model in which $B_i$ was not allowed to vary by reward:

$$P_{PREF}(C) = 1 \, / \left(1 \, + \, \hat{e}(-((B_0)((k_0 + \alpha k_1 + \beta k_2)C + \beta) + x_0 + \alpha x_1))\right). \quad (3c)$$

Model fitting was performed as described in previous sections. We report uncorrected p values; however, all statistically significant results survive Bonferroni correction for multiple comparisons across the two animals (where relevant). Quantile–quantile comparisons between the fit of this decision model and the cumulative Gaussian model described in the previous section indicate a reasonable correspondence between model fits (*Figure 8—figure supplement 3*).

## Site-by-site quantification of effect of reward on microstimulation shift

At each microstimulation site, trials were divided according to whether the available reward size for a correct choice in the trial was large or small. The size of the horizontal shift of the psychometric function induced by electrical microstimulation was re-calculated as described in the previous section, separately for large and small reward trials. To normalise the effect of microstimulation across sites with different associated threshold due to stimulus eccentricity (see *Figure 8*), the horizontal shift of the psychometric function in each reward condition was divided by the psychometric threshold for that reward condition that is, the standard deviation (s.d.) of the fitted cumulative Gaussian (*Equation 1a*; *Uka and DeAngelis 2006*). A Wilcoxon sign-rank test across sites ($p < 0.05$, two-sided) was used to ascertain whether there was an overall significant difference in the size of the microstimulation shift between the large and small reward conditions. A non-parametric test was used because although the normalized shifts were normally distributed (Lilliefors test, $p > 0.05$), the underlying distribution of raw shift values was not normally distributed (Lilliefors test, $p = 0.001$). We report uncorrected p values; however, all statistically significant results survive Bonferroni correction for multiple comparisons across the two animals (where relevant).

## Visual perturbation (△dx) control experiment

We performed a control experiment on one monkey, Ica, in which we used a visual stimulus to mimic the psychophysical effect of microstimulation. We matched stimulus and task parameters to the micro-stimulation experiment but no electrical microstimulation took place. Instead, we added a signal of +0.005° disparity, termed '△dx', to the stimulus in half of the trials, pseudo-randomly selected (see *Salzman et al. 1992*; *Fetsch et al. 2014* for a similar manipulation in a visual motion task). This extra signal was not included in the determination of the correct response, that is, we rewarded the monkey's perceptual choices according to the stimulus disparity without the addition of △dx.

The △dx trials were overtly cued by a change in the colour of the cylinder stimulus, from 100% white dots to 100% black dots (*Figure 7A,B*). This allowed us to explore the behavioural strategy adopted by the animal when trials with and without an additional, non-rewarded visual disparity signal are

clearly cued. We kept reward schedules the same. Therefore, for maximal reward accumulation, the monkey's optimal strategy would to bias its perceptual reports away from the fake signal cylinder disparity on the trials that were overtly cued as Δdx.

Over 5 days, 15 experimental sessions of psychophysical data was collected, with all experimental parameters remaining constant. There were approximately 1000 to 2000 trials in each experimental session. By contrast, there were on average 500 trials in each experimental block at the electrical microstimulation sites. The Δdx control experiment comprised 12,149 trials in total, whilst the Fle and Ica electrical microstimulation experiments comprised 9,411 and 10,592 trials in total, respectively, over all microstimulation sites. Since our objective was to ascertain whether the animal learns to treat the Δdx trials differently from non-Δdx trials, we kept the direction of the Δdx signal the same throughout the control experiment. If the animal fails to adjust its criterion on a reward-size basis in this situation when it is clearly apparent when the additional visual disparity signal is added, it seems very unlikely that they could do so when the signal introduced by microstimulation varied between experiments (as it did for the real microstimulation).

## Time-series analysis of performance fluctuation

Performance accuracy (proportion of correct choices) was calculated over a sliding time window 30 trials wide. To estimate the amount of performance fluctuation over the microstimulation session at a given site, the area between the smoothed performance curve and the horizontal line indicating the mean overall performance over that session was calculated using trapezoidal numerical integration implemented by MATLAB's *trapz* function (see *Figure 8—figure supplement 1A*). Since different sites had microstimulation sessions of different lengths, this area was normalised across sites by division by the total number of trials in the microstimulation session at each site. 'Good' and 'bad' performance epochs were identified as stretches of smoothed performance that were above or below mean site performance, respectively, for at least the length of the smoothing time window (i.e. at least 30 trials). To evaluate whether the reward effect on microstimulation remained within good and bad epochs considered individually, trials from good and bad epochs were pooled across sites and animals and fit with cumulative Gaussians to test whether there was significant interaction between reward and microstimulation shift ($\chi^2$-test of nested models *Equations 2a,c*, p < 0.05, as described in previous sections).

## Acknowledgements

NC held an Usher Cunningham Studentship (DPAG and Exeter College, Oxford). KK is a Royal Society University Research Fellow. This work was supported by the Wellcome Trust and National Institutes of Health, Bethesda, MD.

## Additional information

### Funding

| Funder | Grant reference | Author |
| --- | --- | --- |
| Wellcome Trust | 065511/Z/01/Z and WT101092MA | Andrew J Parker, Kristine Krug |
| Usher Cunningham Studentship (DPAG and Exeter College, Oxford) | Graduate Studentship | Nela Cicmil |
| National Institutes of Health (NIH) | EY000404 | Bruce G Cumming |

The funders had no role in study design, data collection and interpretation, or the decision to submit the work for publication.

### Author contributions

NC, BGC, AJP, KK, Conception and design, Acquisition of data, Analysis and interpretation of data, Drafting or revising the article

## Ethics

Animal experimentation: Animal experimentation was conducted at two locations: University of Oxford, UK, and National Institutes of Health (NIH), Bethesda, MD, United States. At Oxford, all procedures were approved by the United Kingdom Home Office, and strictly complied with the restrictions and provisions contained in the Animals (Scientific Procedures) Act of 1986. At NIH, all procedures strictly complied with US Public Health Service policy on the humane care and use of animals, and the protocol was approved by the National Eye Institute (NEI) Animal Care and Use Committee (protocol #NEI-567). Every effort was made to minimise potential sources of pain, suffering, distress or lasting harm to the animals involved in the study.

## Additional files

### Supplementary file

• Supplementary file 1. Large and small reward conditions do not differ in the proportion of microstimulated trials, nor in the distribution of stimulus disparities. The nature of the reward contingencies in this experiment meant that the reward size for any particular trial was dictated by the monkey's accumulated performance on the previous trials. Therefore, the distribution stimulus disparities and the proportion of microstimulated trials were not *a priori* balanced between large and small reward conditions. Any systematic difference between the distributions of these factors in the two reward conditions might contribute to the reported effect of reward on microstimulation shift. We compared the mean absolute disparity and the percentage of microstimulated trials in small reward versus large reward trials. We found no significant difference in the distribution of stimulus disparities between the reward conditions in either monkey across all sites (Wilcoxon sign-rank test, Fle: n = 20, p = 0.263; Ica: n = 28, p = 0.187) nor when significant PREF sites are considered alone (Fle: n = 13, p = 0.127; Ica: n = 15, p = 0.169). We also found no significant difference in the proportion of microstimulated to non-stimulated trials in the two reward conditions in either monkey across all sites (Wilcoxon sign-rank test, Fle: n = 20, p = 0.737; Ica: n = 28, p = 0.075), nor when significant PREF microstimulation sites were considered alone (Fle: n = 13, p = 0.735; Ica: n = 15, p = 0.064).

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
