## [Decision Letter]

Thank you for submitting your work entitled “Reward modulates the effect of visual cortical microstimulation on perceptual decisions” for peer review at *eLife*. Your submission has been evaluated by Eve Marder (Senior Editor) and four reviewers, one of whom, Matteo Carandini, is a member of our Board of Reviewing Editors. Their opinion is favorable, provided that you can address some concerns, which we hope are addressable with existing data and new analyses.

The reviewers have discussed the reviews with one another and the Reviewing Editor has drafted this decision to help you prepare a revised submission.

This interesting study investigates an important topic that should be of broad general interest: the interaction of expected reward with sensitivity of perceptual choices. The paper shows that there is such an interaction both in choices made purely from sensory inputs and from choices influenced by microstimulation of sensory cortex. When the expected rewards are higher, responses are more veridical and less likely to be swayed by microstimulation.

However, as outlined below, there are concerns about the interpretation of the data, the analysis, and the organization of the paper. First, because of the experimental design, it is not clear whether the effects are due to differences in expected reward or to slower variations in task engagement. Second, it is not clear where the effect takes place: at the level of sensory representations or in the manner that decision processes weigh those representations? Third, microstimulation may have engaged specific strategies that are not the same as during normal sensory stimulation. Fourth, there are some unexplained trends in the data that need further exploration and explanation. Finally, the paper would be more effective if it was more explicit about the specific question that the experiments address.

1) Expected reward or to task engagement?

In the experiments, reward size depended on animal's performance in the preceding trials, increasing gradually when animal performed correctly over consecutive trials. Indeed, the animal received larger rewards only after sequences of correct trials, which are more likely when the animal is more engaged. Does higher reward cause the animals to work harder, or does the reward go up because the animal is working harder? Perhaps, rather than expected reward size, the relevant factor are slower fluctuations in task engagement. In this interpretation expected reward size is not the cause of the observed effects. Rather, it is an incidental factor entirely due to experimental design. Even without manipulations of reward size, we know that non-human primates go through fluctuations in their performance, presumably reflecting their attentional or motivational states.

The ideal way to rule this out would of course be with other experiments where reward size varies independently of performance, and the animal knows it in advance, either through blocking or through a cue such as the color of the fixation spot, which could vary from trial to trial. It would have also been much better if the task allowed for the measurement of reaction times, as these could provide evidence that the animals successfully inferred the trial-by-trial reward size (one could regress response times on the size of the upcoming reward).

In the absence of new experiments, it might be possible to address this concern with some time-series analysis of the present data. One might be able to argue that variations in task engagement would be slower than valuations of upcoming reward.

2) Where does the effect take place?

The paper argues that reward influences perceptual sensitivity, operating “prior to integration of sensory evidence”. However, it is not clear how much prior to that integration. There seem to be two possibilities. One is that “reward affects sensory representations of the stimulus”, i.e. the responses of MT neurons. Another is that it influences their interpretation, i.e. the weight that decision mechanisms assign to sensory evidence in MT. It is not clear which of these two is supported by the paper. It should be able to test the first hypothesis by looking at the firing rate of MT neurons, which the authors presumably have recorded. Does it depend on reward? If it doesn't, the microstimulation data clearly support the second hypothesis. If it does, then it is not clear what the microstimulation data add to the story. The authors presumably have data from MT responses (at least multiunit activity or LFP) to support or falsify these possibilities. The paper would be greatly strengthened by an analysis of such data.

3) Is there something special about microstimulation?

The paper shows that microstimulation is more effective when reward sizes are small, and less effective when they are large. Perhaps this is because the animal is better able to screen out the effects of microstimulation on some trials than others (for some unknown reason). The monkey would therefore be more likely to do better on these trials, and thus reward size would go up. Differences between the Δdx control and the main data limit the value of this control for this concern. In other words, noisy or odd/aberrant inputs from sensory neurons (e.g. electrically evoked responses) might be weighed less compared to the more “normal” inputs (e.g. visually evoked responses of non-stimulated MT neurons). These weights can then change based on expected reward or task engagement. It is not clear that there is any analysis that can address this concern, but perhaps it would help to see it be acknowledged and discussed.

4) Effect of reward on model parameters:

There is a concern about the central conclusion about the effect of reward size on the parameters of the bounded accumulation model, *B* and *k*. It would help to have a schematic figure showing what would happen to the psychometric functions when *k* changes, *B* changes, and both change. The critical passage in the Discussion (“Even though we employed a fixed viewing duration task, differential effects of reward on parameters *k* (stimulus scaling sensitivity) and *B* (distance-to-bound) could be constrained in our model due to the insertion of the microstimulation signal in visual area V5/MT”) is too cryptic to help. In the data analysis, it is also not clear how well the two effects can be separated. While the statistical testing of each parameter seems appropriate, one can't readily see the magnitudes of the effect of freezing each on the overall goodness of fit of the model. This could be solved by adding such information to Table 2.

5) Unexplored trends in the data:

There's a puzzle in that the bulk of the cases where the reward manipulation had a big effect were cases where the microstimulation effect was weak and the performance was good. This can be gleaned by comparing where the bigger effects fell in Figure 3. The only way one can get a big relative effect from a small absolute one is if the slope is quite steep. Why is there no effect of reward for big (in absolute terms) microstimulation effects? Perhaps these are high eccentricity sites where thresholds are high? But then, why would the same effects not be seen in those cases?

The behavioral effect of reward (both in normal trials and in microstimulated trials) is considerably cleaner in animal Ica compared to the animal Fle. Is there any reason for such a difference? Can the analysis be strengthened by considering individual sessions, and statistics across sessions?

Also, it should be possible to improve the analysis by treating reward size as a continuous rather than categorical manner, or at least a different categorization. For instance, it would be helpful to see whether the results hold if trials were categorized to less-than-average and more-than-average reward sizes. For this analysis, the average reward size per trial would be measured for each session and then trials could be separated based on the measured average.

6) Reorganizing the paper around a central question:

This paper is organized in a manner that is appropriate to specialists, but fails to make a coherent case that is understandable and interesting to a wider audience. In the Abstract and Introduction, it repeatedly indicates that it wants to make some measurement simply because it has not been made before. This is insufficient motivation: the fact that something has not been done before does not mean that it is interesting and that it answers some question about brain function. It is much better to start by posing a clear question about brain function, and to organize the paper around that question.

In this paper, the central question seems to be the one posed in the second paragraph of the Introduction: “how information about the value associated with different choices is combined with sensory information to influence the final decision”. In that respect, it might make sense to invert the order of presentation of the data, and start from the results obtained without microstimulation.

For example, a suggested sequence of arguments is: (1) Start from the existing knowledge (e.g. [57] and others) and support it by documenting the effect of reward on psychometric data obtained without microstimulation (the black curves in Figure 5). (2) Model this effect with the bounded accumulation model (Figure 5), and indicate which parameters change. (3) Ask: how does reward influence perceptual sensitivity? One possibility is that it influences the stimulus representation itself, i.e. the responses of MT neurons. Another possibility is that it influences the weight that the decision mechanisms assign to MT responses. (4) See if we can rule out the first possibility by looking at the firing rate of MT neurons. Does it depend on reward? (5) Support the second possibility with the microstimulation data. Show the current Figure 2 and Figure 4 (the red curves), with fits by the drift diffusion model (no need to show the same data twice as currently appear in Figure 4 and Figure 5). Show statistics to support the claim (e.g. Figure 3). This is the key piece of evidence, and there is already sufficient level of analysis in the paper to support it. (6) Show the control, the simulated microstimulation. This will assuage some concerns about the microstimulation and its interpretation.

[Editors' note: further revisions were requested prior to acceptance, as described below.]

Thank you for resubmitting your work entitled “Reward modulates the effect of visual cortical microstimulation on perceptual decisions” for further consideration at *eLife*. Your revised article has been favorably evaluated by Eve Marder (Senior Editor) and a Reviewing Editor. The manuscript has been improved but there are some remaining minor issues that need to be addressed before acceptance, as outlined below.

The paper has much improved in terms of clarity and general organization. The new analyses are very useful, and the new Figure 4 makes a much clearer case. There are, however, some details that still need to be addressed before the paper is ready for publication.

1) The analysis in Figure 3 is still hard to evaluate because the reward schedule depended on performance. Therefore, large rewards were mainly delivered when performance was high. Doesn't this design basically predict the result? If so, please acknowledge this flaw in Results, and discuss it in Discussion.

2) The paper goes back and forth between two conclusions: (**A**) reward affects sensory representations of the stimulus, prior to integration stage (last sentence of Abstract), and (**B**) reward affects both sensory representation and integration stage (last sentence of introduction). In fact, the results support the second conclusion (reward affects both model parameters, *B* and *k*) so the Abstract needs to be edited to reflect this.

3) In fact, the Abstract needs a thorough editing. It is not enough to say that something is not understood in general terms. Specifically, what aspect of it is not understood? And what does the paper do to understand it? And what conclusion can be drawn from the results?

4) The paper often refers to a “sensorimotor stage”. This is presumably meant to indicate LIP? If so, it would be useful to indicate this in Figure 1. Or better, use a different word (decision stage? integration stage?) and again make sure this choice is consistent with the words used in Figure 1.

---

## [Author Response]

1) Expected reward or to task engagement?

In the experiments, reward size depended on animal's performance in the preceding trials, increasing gradually when animal performed correctly over consecutive trials. Indeed, the animal received larger rewards only after sequences of correct trials, which are more likely when the animal is more engaged. Does higher reward cause the animals to work harder, or does the reward go up because the animal is working harder? Perhaps, rather than expected reward size, the relevant factor are slower fluctuations in task engagement. In this interpretation expected reward size is not the cause of the observed effects. Rather, it is an incidental factor entirely due to experimental design. Even without manipulations of reward size, we know that non-human primates go through fluctuations in their performance, presumably reflecting their attentional or motivational states.

The ideal way to rule this out would of course be with other experiments where reward size varies independently of performance, and the animal knows it in advance, either through blocking or through a cue such as the color of the fixation spot, which could vary from trial to trial. It would have also been much better if the task allowed for the measurement of reaction times, as these could provide evidence that the animals successfully inferred the trial-by-trial reward size (one could regress response times on the size of the upcoming reward).

In the absence of new experiments, it might be possible to address this concern with some time-series analysis of the present data. One might be able to argue that variations in task engagement would be slower than valuations of upcoming reward.

Thank you for highlighting this important point and for suggesting that a time series analysis of slower fluctuations in task engagement could help to address this concern. Variations in task engagement may be slower, by an order of magnitude, than the valuations of upcoming reward, as task engagement may be expected to vary over tens of trials whilst valuations of upcoming reward changes over one or two trials under our reward schedule. We now include an additional control analysis figure (Figure 8—figure supplement 1) and discuss the results of these analyses in the Results (in the subsection “Controls: Stimulus eccentricity and slower fluctuations in task engagement”) and Discussion (in the subsection “Controls for animal strategy and performance fluctuation”).

First, we measured the extent of fluctuation in smoothed performance accuracy (a measure of task engagement) over the microstimulation session at each site, by calculating a sliding average of performance accuracy over time windows of 30 trials. The area between this curve and the mean performance accuracy is a measure of performance fluctuation over the microstimulation session: if performance stayed close to average all the way through the session this area would be very small, whereas if performance fluctuated greatly this area would be large as the smoothed performance line moves away from the mean. If the reported reward effect on microstimulation in fact resulted from slower fluctuations in task engagement, with differential microstimulation effects associated with good and bad periods of task engagement, then the size of the reward effect should correlate with the amount of performance fluctuation, i.e. when there is little or no fluctuation, there should be little or no modulation of the microstimulation effect by trial-by-trial reward size. However, we found no correlation between reward effect and performance fluctuation for either animal considered separately or together (p >> 0.05 in all cases). Therefore, we find no evidence that slower fluctuations in task engagement underlie the reward-microstimulation effect.

Secondly, we separated trials into “good” and “bad” task engagement epochs depending upon whether they were located in a (>30 trial) time-window of performance accuracy that was either higher or lower, respectively, than the overall average performance accuracy over the microstimulation session at each site. Separately for trials from good and bad epochs, pooled over sites and animals, we analysed whether the reward effect on microstimulation remained. The reward effect on microstimulation shift remained within both the good and the bad epochs (p < 0.05). Since the effect of reward on microstimulation does not disappear when good and bad performance epochs are separately interrogated, this suggests that fluctuations between good and bad epochs of task engagement cannot be the source of the interactions we found between reward size and microstimulation effect and therefore cannot fully explain our results.

When a time window of 15 trials was taken for these analyses, instead of 30 trials, none of the statistical tests were affected (data not presented in the manuscript).

2) Where does the effect take place?

The paper argues that reward influences perceptual sensitivity, operating “prior to integration of sensory evidence”. However, it is not clear how much prior to that integration. There seem to be two possibilities. One is that “reward affects sensory representations of the stimulus”, i.e. the responses of MT neurons. Another is that it influences their interpretation, i.e. the weight that decision mechanisms assign to sensory evidence in MT. It is not clear which of these two is supported by the paper. It should be able to test the first hypothesis by looking at the firing rate of MT neurons, which the authors presumably have recorded. Does it depend on reward? If it doesn't, the microstimulation data clearly support the second hypothesis. If it does, then it is not clear what the microstimulation data add to the story. The authors presumably have data from MT responses (at least multiunit activity or LFP) to support or falsify these possibilities. The paper would be greatly strengthened by an analysis of such data.

Thank you for raising the point that reward could influence weighting of sensory evidence by decision-making structures, and that it is difficult to differentiate this possibility from the potential effect on sensory representations themselves. We present a new Figure in the Introduction that illustrates where the microstimulation signal is inserted and where reward effects might take place, highlighting where the visually evoked and electrically evoked signals might be combined and where they might still be separate. This taken together with Figure 4 – now in the main part of the Results – lays out more clearly the rationale for the application of the bounded evidence-accumulation model to the behavioural data and the specific conclusions that can be drawn from the combination of microstimulation and computational modelling.

Unfortunately we are not able to report any conclusive results from MU firing rate recordings. Of course, it was not possible to record MU firing rates on the trials where we microstimulated because we cannot microstimulate and record from the same electrode. It is of limited value to record on interleaved non-microstimulation trials due to the substantial changes in MU signal that develop during microstimulation (with large drops in overall rate), and in most cases we do not have the relevant data from these blocks. We have very small datasets of MU recordings at each stimulation site, taken during the baseline psychophysics session immediately prior to the microstimulation experiment. However, because MU is a very noisy measure, and the level of cutting of MU data varied from site to site, our data are simply not enough to be conclusive.

Instead, we acknowledge the possibility of changing weights in the Discussion and discuss it in detail (in the subsection “Limitations of a simplified bounded evidence-accumulation model”), acknowledging that systematic electrophysiological recording from neurons in area V5/MT would be necessary to directly answer this question. Citing recent studies (e.g. [21]), we present an argument that there is currently little evidence to suggest that electrically evoked signals can be differentially weighted by the brain compared to visually evoked signals. Our behavioural data show that – rather than being an aberrant signal – microstimulation seems to be a clear signal that effectively influenced perceptual choices in both small and large reward trials, but the signal’s efficacy is modulated. We also highlight our ∆dx control experiment, which controls for the situation in which microstimulation might produce a different percept that the animal could discount more effectively when the reward is high (in the aforementioned subsection. Please also see our response to point 3 below).

We also make explicit the unique value of a microstimulation intervention to this question (in the same subsection): since microstimulation is a causal intervention, it shows that any effect of reward on the sensory representations is actually directly relevant to the perceptual decision. Alternatively, it could be that any potential changes observed in neuronal firing in V5/MT by reward are mere reflections of some top-down signal, and do not actually affect the perceptual choice – this possibility is ruled out by our results because the relative contributions of visually evoked and electrically evoked signals on the perceptual choice are affected by reward.

3) Is there something special about microstimulation?

*The paper shows that microstimulation is more effective when reward sizes are small, and less effective when they are large. Perhaps this is because the animal is better able to screen out the effects of microstimulation on some trials than others (for some unknown reason). The monkey would therefore be more likely to do better on these trials, and thus reward size would go up. Differences between the Δdx control and the main data limit the value of this control for this concern. In other words, noisy or odd/aberrant inputs from sensory neurons (e.g. electrically evoked responses) might be weighed less compared to the more “normal” inputs (e.g. visually evoked responses of non-stimulated MT neurons). These weights* can *then change based on expected reward or task engagement. It is not clear that there is any analysis that* can *address this concern, but perhaps it would help to see it be acknowledged and discussed.*

Thank you for this comment. We have expanded and clarified our discussion of the ∆dx control to address these points (in the Results, subsection “Control: Visual perturbation to test an adjustment strategy” and in the Discussion, subsections “Controls for animal strategy and performance fluctuation” and “Limitations of a simplified bounded evidence-accumulation model”). The ∆dx experiment can control for the situation in which microstimulation is somehow detectable by producing some different percept that the animal can discount more effectively when reward is high. In the ∆dx experiment, rewards were given for correct choices about the cylinder stimulus with respect to the stimulus disparity before addition of the additional ∆dx disparity signal. Trials with this additional disparity – disparity that had to be discounted for maximal reward – were clearly signalled by a change in colour of the cylinder stimulus from black to white. We now illustrate this in additional panel inserts in Figure 7. Importantly, the effect of reward on the slopes of the psychometric functions (fitted with cumulative Gaussians) is present in the ∆dx control dataset (p = 0.00072), emphasizing similarity between the ∆dx control and the main microstimulation data. We now report this in the Figure 7 legend and in the Results. Since the effect of the ∆dx on perceptual choices did not differ by reward condition, this provides no evidence for the view that the animal could effectively adjust decisions against the microstimulation bias more on large reward trials, had it been aware which were microstimulated trials.

However, we acknowledge in the new Discussion that the ∆dx experiment cannot control for any possible selective weighting of odd/aberrant inputs from electrically evoked sensory neuron, because we added the additional disparity with visual stimulation which by definition is “normal” for these cells (in the subsection “Limitations of a simplified bounded evidence-accumulation model”, second paragraph). Instead, we present an argument that, although it is possible that visually and electrically evoked sensory evidence is differentially weighed in decision-making structures according to reward, we believe that this is unlikely based upon the current evidence (in the same subsection. Please also see our response to point 2 above).

4) Effect of reward on model parameters:

*There is a concern about the central conclusion about the effect of reward size on the parameters of the bounded accumulation model,* B *and* k*. It would help to have a schematic figure showing what would happen to the psychometric functions when* k *changes,* B *changes, and both change. The critical passage in the Discussion (“Even though we employed a fixed viewing duration task, differential effects of reward on parameters* k *(stimulus scaling sensitivity) and* B *(distance-to-bound) could be constrained in our model due to the insertion of the microstimulation signal in visual area V5/MT”) is too cryptic to help. In the data analysis, it is also not clear how well the two effects* can *be separated. While the statistical testing of each parameter seems appropriate, one* can*'t readily see the magnitudes of the effect of freezing each on the overall goodness of fit of the model. This could be solved by adding such information to Table 2.*

Thank you for pointing out that we did not present the effect of reward on model parameters clearly enough. We agree that it is very helpful for the main paper to have a schematic figure showing what would happen to the psychometric functions when *k* changes and *B* changes and we have included new figure panels, Figure 4, to show this. This Figure illustrates the predictions that the model makes regarding how each parameter affects the microstimulation shift. Specifically, we explain (in the subsection “Bounded accumulation can model interactions between reward and perceptual decision-making”) how both parameters can improve visual discrimination performance, seen as a steepening of the psychometric functions, but only parameter *k* (stimulus sensitivity) is associated with a concomitant decrease in microstimulation shift whilst parameter *B* does not affect microstimulation shift. This is how the microstimulation intervention can separate the effects of the two parameters *k* and ***B***. We hope that this explanation is no longer cryptic.

We have clarified in the Methods (in the subsection “Decision model analysis: bounded evidence accumulation”) and throughout the Results that in all cases we used a χ^2^ likelihood-ratio test to statistically evaluate whether the full model, in which all parameters were allowed to vary by reward, fitted the data better than a nested model in which only the parameter in question was not allowed to vary by reward. In Table 1, we report the size of the change of parameter estimates by reward in the best-fitting model (which was the full model in all cases), and we now also report the negative log likelihood of the fits of the two models (the measure of goodness of fit used to estimate parameter values) and the resultant chi value used to calculate the p value for model comparisons.

We have added this information to Table 1 and have removed Table 2 from the manuscript because it did not add to the Results. Instead, we state in the text that the results of model fitting were not affected by the inclusion of all microstimulation sites rather than just the significant PREF sites.

5) Unexplored trends in the data:

*There's a puzzle in that the bulk of the cases where the reward manipulation had a big effect were cases where the microstimulation effect was weak and the performance was good. This* can *be gleaned by comparing where the bigger effects fell in*
Figure 3*. The only way one* can *get a big relative effect from a small absolute one is if the slope is quite steep. Why is there no effect of reward for big (in absolute terms) microstimulation effects? Perhaps these are high eccentricity sites where thresholds are high? But then, why would the same effects not be seen in those cases?*

We apologise for the confusion caused by the log scale in this Figure. We have edited the axes of this figure to clarify that it is on a log scale (now Figure 6). We have explored the suggested trends and include a new figure (Figure 8) to lay out the relationships between reward effect, raw overall microstimulation effect, site eccentricity and perceptual thresholds. As expected, the cylinder discrimination threshold is significantly correlated with site eccentricity (Figure 8) and the raw overall microstimulation effect is significantly correlated with the threshold at a given site (Figure 8). However, interestingly, the raw effect of reward on microstimulation shift is, if anything, bigger for high eccentricity sites (where thresholds are high) and for large raw overall microstimulation shift effects (significant for one monkey, Ica, and for both monkeys considered together, but not significant for monkey Fle) (Figure 8). These correlations disappear entirely when the effect of reward is considered over the normalised (by division by threshold) microstimulation shifts, showing that in relation to threshold, the effect of reward manipulation is in fact constant.

Thank you very much for suggesting that these trends are followed up. We think that this analysis (Figure 8) adds greatly to the paper in explaining that in relation to threshold, the reward effect is consistent over site eccentricity and raw overall microstimulation effect.

The behavioral effect of reward (both in normal trials and in microstimulated trials) is considerably cleaner in animal Ica compared to the animal Fle. Is there any reason for such a difference? Can the analysis be strengthened by considering individual sessions, and statistics across sessions?

Thank you for this suggestion. We highlight Figure 6, which inspects the effect of reward on microstimulation shift size in an independent, site-by-site analysis i.e. across microstimulation sessions, for each animal separately. In all cases, the microstimulation effect was significantly smaller for large reward trials compared to small reward trials, fully supporting the results from the pooled data and decision model fitting. In Figure 8—figure supplement 1, we analyse the effect of reward in good and bad performance epochs within microstimulation sessions, with the same result (please also see our response to point 1).

Although pooled results for monkey Ica may seem somewhat clearer, the data are also somewhat noisier than for Fle, because Ica was run with fewer trials. It is difficult for us to speculate on the individual differences between the two animals, which were tested at different institutions. The important point is that for both animals, psychometric functions were steeper (performance improved) in large reward trials, and the interaction between reward and microstimulation shift was highly significant for each animal separately whether considered site-by-site or in the data pooled across sites. In particular, the horizontal shift between the psychometric functions due to microstimulation is affected by the reward condition for both animals, regardless of other individual differences between the performance of the animals.

Also, it should be possible to improve the analysis by treating reward size as a continuous rather than categorical manner, or at least a different categorization. For instance, it would be helpful to see whether the results hold if trials were categorized to less-than-average and more-than-average reward sizes. For this analysis, the average reward size per trial would be measured for each session and then trials could be separated based on the measured average.

Thank you for this suggestion. In a sense, our current categorisation of reward size already combines rewards that are up to and including the average reward size across the three reward steps under the “small” reward condition, and rewards that are larger than the average in the “large” reward condition (in Figure 2 and also in Methods, in the subsection “Reward Schedule”). We recognise the potential value in treating reward size in a continuous manner and testing different categorisations. However, since there are only three reward categories in our dataset, small, medium and large (see Figure 2 of reward schedule) – with small and median combined under the overall “small” reward condition in all of the main analysis – the only way to change categorisation of reward size would be to include the medium reward trials with the large reward category instead of the small or to exclude those trials entirely. Since there are fewer trials for the medium reward category and small reward categories relative to the large reward categories in any case, this would not affect the large reward category results much but would leave the small reward category with half to two-thirds of its current trial number, increasing noise. A previous analysis that excluded medium reward size trials showed the same statistical results on the main effect (data not presented in the manuscript).

6) Reorganizing the paper around a central question:

This paper is organized in a manner that is appropriate to specialists, but fails to make a coherent case that is understandable and interesting to a wider audience. In the Abstract and Introduction, it repeatedly indicates that it wants to make some measurement simply because it has not been made before. This is insufficient motivation: the fact that something has not been done before does not mean that it is interesting and that it answers some question about brain function. It is much better to start by posing a clear question about brain function, and to organize the paper around that question.

In this paper, the central question seems to be the one posed in the second paragraph of the Introduction: “how information about the value associated with different choices is combined with sensory information to influence the final decision”. In that respect, it might make sense to invert the order of presentation of the data, and start from the results obtained without microstimulation.

*For example, a suggested sequence of arguments is: (1) Start from the existing knowledge (e.g.*
[57]
*and others) and support it by documenting the effect of reward on psychometric data obtained without microstimulation (the black curves in*
Figure 5*). (2) Model this effect with the bounded accumulation model (*Figure 5*), and indicate which parameters change. (3) Ask: how does reward influence perceptual sensitivity? One possibility is that it influences the stimulus representation itself, i.e. the responses of MT neurons. Another possibility is that it influences the weight that the decision mechanisms assign to MT responses. (4) See if we* can *rule out the first possibility by looking at the firing rate of MT neurons. Does it depend on reward? (5) Support the second possibility with the microstimulation data. Show the current*
Figure 2
*and*
Figure 4
*(the red curves), with fits by the drift diffusion model (no need to show the same data twice as currently appear in*
Figure 4
*and*
Figure 5*). Show statistics to support the claim (e.g.*
Figure 3*). This is the key piece of evidence, and there is already sufficient level of analysis in the paper to support it. (6) Show the control, the simulated microstimulation. This will assuage some concerns about the microstimulation and its interpretation.*

Thank you very much for this detailed suggestion to restructure the paper, which we have adopted and believe greatly clarifies the presentation of our results and conclusions. According to this suggestion, we have rewritten the Introduction around the central question: “how information about the value associated with different choices is combined with sensory information to influence the final decision”. In the Introduction we have also included a new Figure 1 to lay out the context in which reward effects on perceptual decisions is being investigated i.e. the different stages of the perceptual decision pathway in the brain, the location of the inserted microstimulation signal, and the possible locations of reward effects.

We have restructured the Results according to the following: (1) laying out the behavioural evidence of improved performance in large reward trials compared to small reward trials from the non-stimulated datasets collected prior to the microstimulation sessions (Figure 3), (2) explaining how the bounded accumulation model can account for this effect in different ways (parameters *k* and *B*) but that these make different predictions about what should happen to microstimulation in large reward trials i.e. should it be affected (parameter *k*) or not (parameter *B*) (Figure 4), (3) fitting the pooled microstimulation data from each monkey with the bounded accumulation model and showing that reward affects the microstimulation shift for perceptual decisions, implicating parameter *k* (Figure 5), along with relevant statistics, (4) independent site-by-site statistical analysis to confirm that reward modulates the effect of cortical microstimulation on perceptual decisions, with a figure supplement to show how this looks at the level of a single site (Figure 6), (5) Presentation of the ∆dx control of simulated microstimulation to rule out some alternative interpretations of the effect of reward on microstimulation, (6) Figure 8 and three figure supplements to report the control investigations of additional trends in the data (point 5) and time-series analysis of performance fluctuations (point 1).

We restructured the Discussion to focus on the central question first i.e. implications of our results for how reward signals are combined with sensory information during perceptual decision-making. We discuss our results in the context of previous studies and suggest future experiments. We added new headings to the Discussion to deal with the other points brought up by the reviewers’ comments.

We were not able to include an analysis for V5/MT MU firing rates by reward because of noisy data and small numbers of trials (please see detailed response to point 2).

[Editors' note: further revisions were requested prior to acceptance, as described below.]

*1) The analysis in*
Figure 3
*is still hard to evaluate because the reward schedule depended on performance. Therefore, large rewards were mainly delivered when performance was high. Doesn't this design basically predict the result? If so, please acknowledge this flaw in Results, and discuss it in Discussion.*

We agree that it is important to clarify this point regarding Figure 3. We have now acknowledged this issue earlier in the paper (in the subsection “Large reward trials are associated with better performance”). If there is no trial-by-trial modulation of performance by available reward size, and no longer-term fluctuations between higher and lower performance states through the session, then even under our experimental design there should be no association between improved performance and large available reward trials. This is because we separated trials according to the size of reward available for a correct choice, rather than whether or not reward was actually delivered (which would only be for correct choices and so by definition would entail perfect performance). If, however, there are fluctuations between good and bad performance states (epochs) throughout the session, then, as pointed out in point 1, there would be an automatic association between better performance and large reward trials, which occur more frequently in the high performance epochs than low performance epochs, as a consequence of our experimental design. (NB. Even then, this would not necessarily entail the main results of our study, i.e. it would not entail that either performance or expected reward size – or some weighted combination of the two – must necessarily affect the effect of microstimulation).

Accordingly, we now acknowledge this potential confound between high performance and large reward trials earlier in the Results (in the aforementioned subsection) and point to the control analysis that shows that such performance fluctuations do not underlie the main results of this study. We also now acknowledge this potential confound in the section of the Discussion (in the subsection “Controls for animal strategy and performance fluctuation”) that pertains to the control analyses for performance fluctuations and the reward-microstimulation effect.

*2) The paper goes back and forth between two conclusions: (****A****) reward affects sensory representations of the stimulus, prior to integration stage (last sentence of Abstract), and (****B****) reward affects both sensory representation and integration stage (last sentence of introduction). In fact, the results support the second conclusion (reward affects both model parameters,* B *and* k*) so the Abstract needs to be edited to reflect this.*

Thank you for bringing this inconsistency to our attention. Although our results show that the effects of reward occur at both stages, we had previously focussed on the effect of reward on sensory representations because that is the novel finding with respect to previous literature. However, we acknowledge that this had been presented in a confusing and inconsistent way. We have now edited the Abstract and main text to consistently reflect conclusion (**B**), i.e. that reward affects both stages.

*3) In fact, the Abstract needs a thorough editing. It is not enough to say that something is not understood in general terms. Specifically, what aspect of it is not understood? And what does the paper do to understand it? And what conclusion* can *be drawn from the results?*

We have thoroughly re-edited the Abstract to specify what aspects of the effect of reward on perceptual decision-making were not understood, and we have carefully checked for and removed typos from the Abstract (see also response to point 2 above).

*4) The paper often refers to a “sensorimotor stage”. This is presumably meant to indicate LIP? If so, it would be useful to indicate this in*
Figure 1*. Or better, use a different word (decision stage? integration stage?) and again make sure this choice is consistent with the words used in*
Figure 1*.*

Thank you for pointing out that way we described this was not consistent. We have decided to now refer only to the “integration stage” of the decision-making pathway (not “sensorimotor stage” or “sensorimotor integration stage”). Where relevant, we have specified that the integration stage is thought to be represented in sensorimotor regions such as area LIP. In line with these changes, we have also edited Figure 1 to specify that area LIP represents the “decision variable (integration stage)”.